

# Sensitivity of Totten Glacier dynamics to sliding parameterizations and ice shelf basal melt rates

Yiliang Ma[1], Liyun Zhao[1], Rupert Gladstone[2], Thomas Zwinger[3], Michael Wolovick[4], John C. Moore[2]

[1]State Key Laboratory of Earth Surface Processes and Resource Ecology, Faculty of Geographical Science, Beijing Normal University, Beijing 100875, China
[2]Arctic Centre, University of Lapland, Rovaniemi, Finland
[3]CSC – IT Center for Science Ltd., Espoo, Finland
[4]Glaciology Section, Alfred Wegener Institute, Bremerhaven, Germany

*Correspondence to*: Liyun Zhao (zhaoliyun@bnu.edu.cn) and John C. Moore (john.moore.bnu@gmail.com)

**Abstract.**

Totten Glacier in East Antarctica holds a sea level potential of 3.85 m and is mostly grounded below sea level. It has the third highest annual ice discharge, 71.4±2.6 Gt yr[-1], among East Antarctic outlet glaciers and has been losing mass over recent decades. Recent thinning of the Totten ice shelf is likely to be due to high basal melt rates driven by increasing intrusion of warm Circumpolar Deep Water. Here we simulate the evolution of the Totten Glacier subregion using a Full-Stokes model with different basal sliding parameterizations (linear Weertman, nonlinear Weertman, and regularised Coulomb) as well as sub-shelf melt rates to quantify their effect on the projections. The modelled grounding line retreat and decline in ice volume above floatation using the linear Weertman and the regularised Coulomb sliding parameterizations are very close, and both larger than that using the nonlinear Weertman sliding parameterization. The simulated grounding line retreat occurs only with maximal basal melt rate higher than 40 m yr[-1], and is mainly on the eastern and southern grounding zone of Totten Glacier. The change of sub-shelf cavity thickness is not sensitive to the choice of basal sliding parameterization, only to sub-shelf melt rates, yielding strong volume above floatation dependence on melting through the mechanism of reduced buttressing.

## 1. Introduction

Totten Glacier (TG), located in the Aurora Basin, contains 3.85 m of sea-level change equivalent, had an annual ice discharge of 71.4±2.6 Gt yr[-1] from the year 2009 to 2017 which is the third highest ice discharge from the East Antarctic Ice Sheet (EAIS), and shows on-going dynamic change (Rignot et al., 2019; Greenbaum et al., 2015). TG speed slowed down from 1989 to 2001, then accelerated and stayed at relatively constant speed over the 2005-2015 period (Li et al., 2016). The rate of TG mass loss increased by 28% to 7.3 Gt yr[-1] between 2003-2017 relative to its 1979-2003 rate (Rignot et al., 2019). Bedrock elevations in the TG subbasin are mostly below sea level, with the deepest region nearly 2000 m below sea level (Morlighem et al., 2020). The grounding line (GL) retreated by 1-3 km during the period 1996-2013, and the ice at the GL



thinned by 0.7 ± 0.1 m (Li et al., 2015). A further GL retreat of 0.5-2 km along the ice plains beneath the glacier central trunk occurred from 2013 to 2021 (Li et al., 2023). The mass imbalance of TG and the existence of deepening topography extending far inland have raised concerns that this subbasin has large and rapid sea level rise potential (Li et al., 2016;

Morlighem et al., 2020).

Warm modified Circumpolar Deep Water (mCDW; Ribeiro et al., 2021) at about 0 °C sits below 500 m depth on the continental shelf. This mCDW invades the ice shelf cavity through a deep channel incised in the sea floor at a rate of 0.22 ± 0.07 Sv (Rintoul et al., 2016; Silvano et al., 2017). This inflow of heat induces rapid basal melting, that may thin the ice shelf,

remove grounding on sea floor topographic high points and hence reduce the ice shelf buttressing force on the grounded ice. Observations suggest that episodic GL retreats are coincident with high thinning rates and high ice velocities (Li et al., 2023), and that TG may be very sensitive to deep ocean temperature (Li et al., 2016).

The future evolution of TG has been studied with various numerical ice sheet models employing different relationships

between mass loss and ocean thermal forcing. Some studies used sub-shelf melt rate parameterizations (Sun et al., 2016; Pelle et al., 2020), and some used coupled ice-ocean modelling (McCormack et al., 2021; Pelle et al., 2021). All these studies project that TG will experience continuous and significant retreat on its eastern sectors under climate warming scenarios. Upstream of the GL, ice flow is mostly controlled by basal sliding and vertical shear stress. In contrast with the many studies on the impact of sub-shelf melt rates on TG dynamics, there has not been an investigation of TG dynamics sensitivity to

basal sliding parameterization.

Several types of basal sliding parameterization have been proposed and improved over the past 70 years (Weertman 1957; Nye 1970; Kamb, 1970; Fowler, 1981; Budd et al., 1979; Schoof, 2005; Gagliardini et al., 2007; Tsai and others, 2015). The Weertman sliding parameterization is formulated assuming that temperate ice slides perfectly on a hard bed, and the

effective pressure (the difference between ice overburden pressure and basal water pressure) is sufficiently high that no cavities form as the ice flows over the bed, so the basal drag does not depend on effective pressure. The linear relation between basal friction and basal sliding velocity is only valid when regelation dominates or under the simplifying assumption that ice deforms like a Newtonian viscous material (Nye 1970; Kamb, 1970; Cuffey and Paterson, 2010). But this linear Weertman sliding is used very widely in ice sheet dynamic simulations mainly due to its simplicity (e.g. Sun et al.,

2016; Nowicki et al., 2016; Pelle et al., 2020). A nonlinear Weertman sliding parametrization with an exponent of 1/3 applies in the lack of large enough obstacle sizes to induce regelation processes, in which case sliding is dominated by ice-deformation on the largest occurring roughness features (Fowler, 1981). In the case of fast sliding on a hard bed, cavity formation increases with sliding rate, the basal drag attains a constant value or even decreases as sliding rate increases, showing a strong dependence on effective pressure (Cuffey and Paterson, 2010). Effective pressure has been included in

several basal sliding parameterizations (Budd et al., 1979, 1984; Schoof, 2005; Tsai and others, 2015; Gagliardini et al.,



2007). In the Budd sliding parameterization (Budd et al., 1979, 1984), basal drag is proportional to both basal sliding velocity and effective pressure, and can reach arbitrarily high values. In the Coulomb sliding parameterization, basal drag is only proportional to effective pressure, and thus has upper limit; this Coulomb plastic rheology is able to describe sliding over tills (Iverson and others, 1998). Tsai et al. (2015) proposed a sliding parameterization which accommodates the

Weertman type friction and the Coulomb friction regimes, and the basal drag instantaneously switches from Coulomb friction at low effective pressure to Weertman type friction at high effective pressure. Schoof (2005) and Gagliardini et al. (2007) developed a regularized Coulomb sliding parameterization, which naturally has a continuous transition between Coulomb friction regime and Weertman type friction regime.

The choice of basal sliding parameterizations used by models can have a significant impact on the simulated GL retreat rate and glacier mass loss (Gladstone et al., 2014, 2017; Brondex et al., 2017, 2019). Brondex et al. (2017) found that simulations on an idealised two-dimensional ice sheet geometry with different basal sliding parameterizations (the nonlinear Weertman with exponent of 1/3, Schoof, and Budd sliding parameterization) show significantly different transient behaviour although they start from identical initial states, and the nonlinear Weertman sliding parameterization forecasts the least volume above

floatation (VAF) loss and Budd sliding parameterization the most. Yu et al. (2018) showed that the use of Budd sliding parameterization produces more VAF loss under warming scenarios than linear Weertman sliding parameterization in simulations of Thwaites Glacier. Brondex et al. (2019) simulated the evolution of Amundsen Sea Embayment for 100 years and found that GL retreat and VAF loss with the linear and nonlinear (with exponent of 1/3) Weertman sliding parameterizations are less than that with a regularised Coulomb sliding parameterization, which are less than that with Budd

sliding parameterization. They also found that simulations using the non-linear versions (with exponent of 1/3) of the Budd and Weertman sliding parameterizations produce more VAF losses than those using the linear versions although they lead to similar or less grounded ice area loss.

In this study, we conduct a series of simulations to assess the dynamic response of TG under different basal sliding

parameterizations and sub-shelf melt rates from the year 2015 to 2050. We use the full-Stokes ice-flow model, Elmer/Ice (Gagliardini et al., 2013). Since Elmer/Ice has more complete physics than the simplified models that have been used in previous TG simulations, we expect to gain more insight into basal processes. We specify three types of basal sliding parameterizations (linear and nonlinear Weertman, and regularised Coulomb) and a range of sub-shelf melt rates with the same parameterization but different maximal basal melt rate of 0, 20, 40, 80 and 160 m yr$^{-1}$. We compare the results from the

different simulations and investigate the impact of different basal sliding parameterizations and sub-shelf melt rates on TG evolution.



## 2. Data, model domain and mesh refinement

### 2.1 Data

The Totten subregion is part of the Aurora basin of the EAIS. We take its 2D domain boundary outline and ice front location
from the MEaSUREs Antarctic Boundaries for IPY 2007-2009 from Satellite Radar, Version 2 dataset (Mouginot et al.,
2017). The bed elevation, surface elevation (Fig. 1) and ice thickness are derived from MEaSUREs BedMachine Antarctica,
version 2, at a resolution of 500 m (Morlighem et al., 2020). The surface ice velocity data (Fig. 1), with a resolution of 450 m,
are obtained from MEaSUREs InSAR-based Antarctic ice velocity Map, version 2 (Rignot and Scheuchl, 2017).

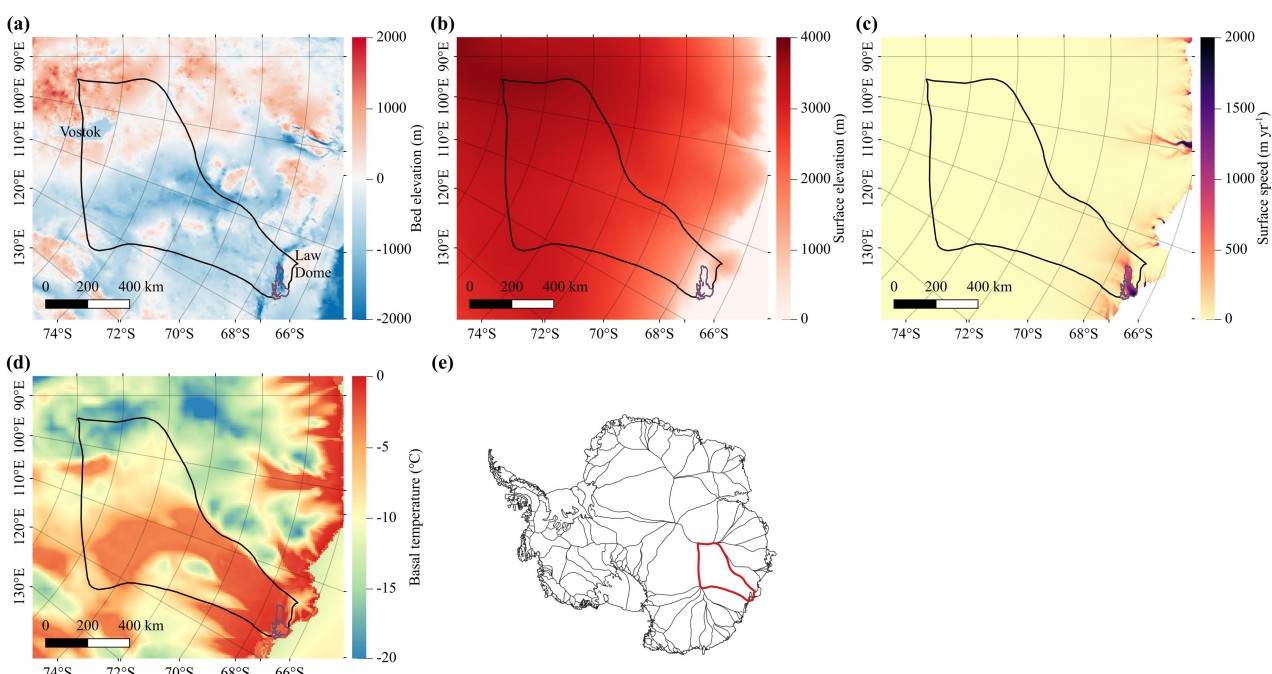

**Figure 1.** Bed elevation **(a)**, surface elevation **(b)**, surface ice flow speed **(c)**, modelled basal temperature **(d)** of Totten
subregion and its surroundings. The bed and surface elevations are from MEaSUREs BedMachine Antarctica, version 2. The
solid black curve is the outline of Totten subregion, from MEaSUREs Antarctic Boundaries for IPY 2007-2009 from
Satellite Radar, Version 2 dataset (Mouginot et al., 2017). The purple curve is the GL and ice front. Plot **(e)** shows drainage
basin divisions and the location of our domain (red curve) in Antarctica. Subglacial lake Vostok and Law Dome are marked
in plot (a). The prescribed ice temperatures are taken from the output of the ice sheet model SICOPOLIS (Greve et al., 2020).

Ice sheet temperature data, at 8 km resolution (Fig. 1d), are sourced from the output files of the ice sheet model SICOPOLIS
(Greve et al., 2020), which is initialized by a paleoclimatic spin-up over 140 000 years until 1990, when the topography is



nudged towards that of the present-day (Greve et al., 2020). The geothermal heat flux applied in their study is from Martos et
      al. (2017), at a resolution of 15 km. The surface mass balance data are obtained from the output of the Community Earth
      System Model Version 2 (CESM2) of the Coupled Model Intercomparison Project Phase 6 (CMIP6) under the SSP5-8.5
      scenario at 32 km resolution (Nowicki et al., 2016). The ocean forcing is simplified to a parameterization scheme based on
      the Whole Antarctic Ocean Model (WAOM v1.0) output for modelling the basal melt rate of the ice shelves (see Section 3.3).

See Table 1 for the dataset details.

**Table 1. Datasets used in this study.**

| Variable name | Dataset | Resolution | Period | Reference |
|---|---|---|---|---|
| Subbasin boundary | MEaSUREs Antarctic Boundaries for IPY 2007-2009 from Satellite Radar, Version 2 | 25-250 m | 1992-2015 | Mouginot et al., 2017 |
| Surface elevation, bedrock elevation, and ice thickness | MEaSUREs BedMachine Antarctica version 2 | 500 m | 1970-2019 | Morlighem, 2020 |
| Surface ice velocity | MEaSUREs InSAR-based Antarctic ice velocity Map, version 2 | 450 m | 1996-2016 | Rignot and Scheuchl, 2017 |
| Ice temperature | Output of SICOPOLIS | 8 km | 2015 | Greve et al., 2020 |
| SMB | CESM2 output under SSP5-8.5 | 32 km | 2015-2050 | Nowicki et al., 2016 |
| ice shelf basal melt rate | Whole Antarctic Ocean Model (WAOM v1.0) output | 2 km | 2007 | Richter et al., 2022 |

## 2.2 Mesh generation and refinement

We use Gmsh software (Geuzaine and Remacle, 2009) to generate an initial 2D horizontal footprint mesh within the domain
      boundary. The initial grid was then optimised by the anisotropic mesh adaptation code Mmg (Dapogny et al., 2014), which
      helps to refine the mesh where ice velocity and thickness gradients identify regions needing increased resolution. The
      resulting mesh is shown in Fig. 2 and has minimum and maximum element sizes of approximately 900 m and 20 km. The 2D
      mesh contains 43 580 elements and 22 236 nodes. It is then vertically extruded using 20 terrain-following layers, with a

vertically refined mesh toward the base.



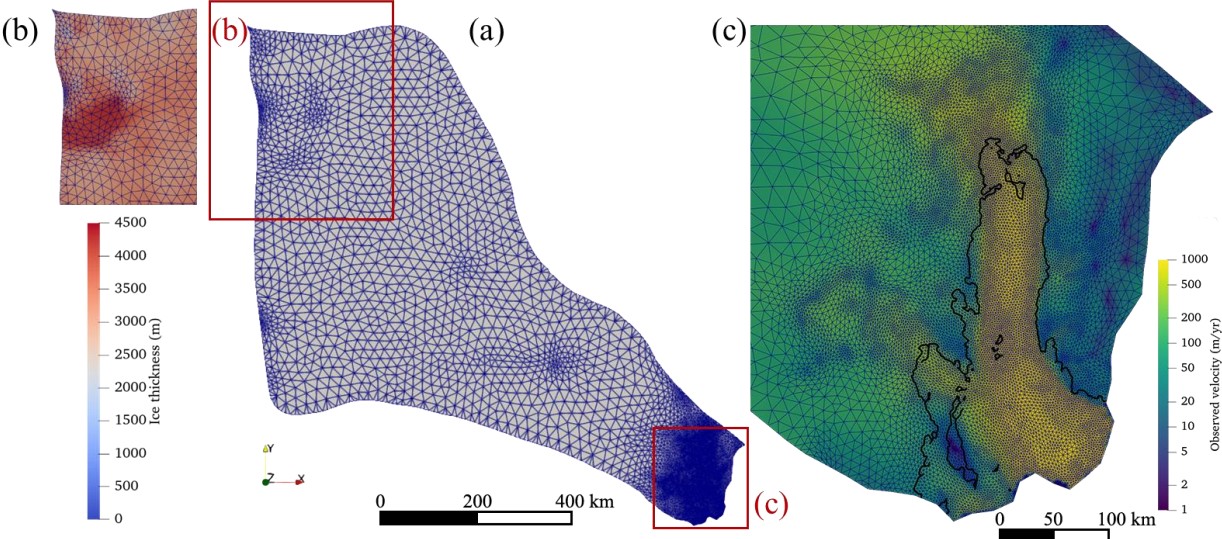

**Figure 2.** The refined 2D horizontal domain footprint mesh **(a)**. Boxes outlined in panel **(a)** are shown in detail and overlain with ice thickness in panel **(b)** and with surface ice velocity in panel **(c)**. The solid black curves in (c) represent the positions of the GL.

## 3. Method

### 3.1 Ice Flow Model

We carried out all the simulations using the ice flow model Elmer/Ice (Gagliardini et al., 2013) with version 9.0. We solve the Stokes equation and use an inverse method to determine basal friction coefficient in the linear Weertman sliding parameterization and the stress enhancement factor. The Stokes equation includes conservation equations for the momentum and mass:

$$\nabla \cdot \boldsymbol{\tau} - \nabla p = \rho_i \boldsymbol{g}, \tag{1}$$

$$\nabla \cdot \boldsymbol{v} = 0, \tag{2}$$

where $\boldsymbol{\tau}$ is the deviatoric stress tensor, $p$ is the isotropic pressure, $\rho_i$ is ice density, $\boldsymbol{g}$ is the acceleration due to gravity (0, 0, −9.81) m s$^{-2}$, and $\boldsymbol{v}$ is ice velocity. The effective viscosity, $\eta$, for the ice-rheology,

$$\boldsymbol{\tau} = 2\eta\dot{\boldsymbol{\varepsilon}}, \tag{3}$$

given by Glen's flow law (Cuffey and Paterson, 2010),

$$\eta = \frac{1}{2}(E \cdot A(T))^{-\frac{1}{n}} \dot{\varepsilon}_e^{\frac{(1-n)}{n}}, \tag{4}$$

is dependent on ice temperature via the flow rate factor $A(T)$, defined by an Arrhenius equation. We use an exponent of $n=3$. $\dot{\boldsymbol{\varepsilon}}$ denotes strain rate and the term $\dot{\varepsilon}_e = \sqrt{tr(\dot{\boldsymbol{\varepsilon}}^2)/2}$ is the effective strain rate. The ice temperature distribution comes from the output of ice sheet model SICOPOLIS (Greve et al., 2020). We introduce $E_\eta = E^{-1/(2n)}$, to obtain



$$\eta = \frac{1}{2} E_\eta^2 A(T)^{-\frac{1}{n}} \dot{\varepsilon}_e^{\frac{1-n}{n}}, \tag{5}$$

which ensures $\eta > 0$ for any value of $E_\eta$. We refer $E_\eta$ to as the 2D vertically constant stress enhancement factor which is set initially to unity everywhere, indicating no stress enhancement, before performing the viscosity inversion. Values of $E_\eta < 1$ indicate softer and faster deforming ice, whereas higher values, $E_\eta > 1$, mean stiffer ice.

The ice surface is assumed to have a vanishing Cauchy-stress vector. At the ice front under the sea level and the bottom of ice shelf, the normal Cauchy-stress is set equal to the hydrostatic water pressure. The so-called sea spring (Gagliardini et al., 2013) stabilization method is applied to the bottom free surface in the case of a temporal evolution away from the hydrostatic equilibrium. On the lateral boundary, the normal stress is equal to the hydrostatic ice pressure and the normal velocity is set to zero, the ice is free to slip in the tangential direction. The bed for grounded ice is assumed to be rigid, impenetrable, and fixed over time. The normal component of the velocity is set to zero at the ice–bed interface. We implement alternative basal sliding relations (Sect. 3.2) for the grounded ice in Elmer/Ice. For transient simulations, we prescribe a surface mass balance at the ice surface (Sect. 2.1), and sub-shelf basal melt rate parameterization along the surface in contact with the ocean (Sect. 3.3).

### 3.2 Basal sliding parameterizations and conversion between them

We use three basal sliding parameterizations in this study: linear Weertman sliding (Eq. (6)), nonlinear Weertman sliding (Eq. (7); Weertman, 1957) and regularised Coulomb sliding parameterization (Eq. (8-9); Schoof, 2005; Gagliardini et al., 2007) given by

$$\boldsymbol{\tau}_b = C_{LW} \boldsymbol{u}_b, \tag{6}$$

$$\boldsymbol{\tau}_b = C_{NW} \boldsymbol{u}_b^{1/m}, \tag{7}$$

$$\boldsymbol{\tau}_b = C_S N \left[ \frac{\boldsymbol{\chi} \cdot \boldsymbol{u}_b^{-m}}{1 + a \cdot \chi^q} \right]^{\frac{1}{m}} \boldsymbol{u}_b, \tag{8}$$

$$\chi = u_b / C_S^m N^m A_S, \tag{9}$$

where $\boldsymbol{\tau}_b$ is the basal shear stress, $\boldsymbol{u}_b$ is the vector of basal sliding velocity, $u_b$ is basal sliding speed, $C_{LW}$ and $C_{NW}$ are friction coefficients of linear and nonlinear Weertman sliding parameterization. We set $m$=3. The regularised Coulomb sliding parameterization considers the existence of water cavities which may be present between ice and soft sediment layers. $A_S$ is the sliding parameter without cavitation and $C_S$ is the maximum value reached by $\tau_b / N$ (Gagliardini et al. 2007). In Eq. (8), $q$ is a post-peak exponent satisfying q≥1; $a = (q-1)^{q-1} / q^q$. We assume $q$=1, obtaining $a$=1. $N$ represents the



effective pressure on the ice bottom and is defined as $N = \sigma_{nn} - p_w$, where $\sigma_{nn}$ is the normal stress at the bed, and $p_w$ is the water pressure.


Lower effective pressure reduces the basal friction and enhances the basal sliding speed. This effect has been included in several sliding parameterizations (Budd et al., 1984; Schoof, 2005; Tsai et al., 2015). Hydrology models determining the effective pressure distribution rely on a set of poorly constrained parameters such as sheet and channel conductivities and basal melt water input that are hard to measure (Werder et al., 2013). Therefore, many authors take effective pressure as the

hydrostatic pressure imposed by an assumed perfect connection between the subglacial drainage system and the ocean (Morlighem et al., 2010; Tsai et al., 2015; Gladstone et al., 2017). For TG, Dow et al. (2020) simulated the basal hydrology using GlaDS subglacial hydrology model to best match the specularity content, and found the modelled water pressure is ~95% of overburden in most of the region. Motivated by their findings, in this study, we take the effective pressure as 5% of overburden. This assumption alters simulation results significantly from those under the assumption of perfect hydrostatic

connection.

We use the same basal shear stress and basal sliding velocity field found from the linear Weertman sliding inversion and convert the friction coefficient obtained during the inversion to use in the other two sliding parameterizations. In Eq. (6), we set $C_{LW} = 10^\beta$ to make sure $C_{LW}$ is positive. The conversion from $C_{LW}$ to $C_{NW}$ then is trivially given by $C_{NW} = 10^\beta u_b^{1-1/m}$. To

convert the friction coefficient from a linear Weertman to a regularised Coulomb sliding parameterization, as given in Eq. (8), we rearrange the latter

$$\left( \frac{\tau_b}{C_S N} \right)^m = \frac{\chi}{1 + a \cdot \chi^q}, \tag{10}$$

The limit forms of Eq. (10) with small $\chi \to 0$ or large $\chi \gg 1$ correspond to slow-flowing regimes with large effective pressure and fast-flowing regimes with small effective pressure, respectively.

In the limit of small $\chi$, we recover a non-linear Weertman law

$$\tau_b = A_S^{-1/m} u_b^{1/m}. \tag{11}$$

Combining Eq. (11) and Eq. (6) with $C_{LW} = 10^\beta$, we get

$$A_s = (10^\beta \cdot u_b^{1-1/m})^{-m}. \tag{12}$$

In the limit of Eq. (10) for large $\chi$ (i.e., $u_b$ is large and $N$ is relatively small), we obtain a Coulomb-type of friction

$$\tau_b = C_s N. \tag{13}$$



We modify Eq. (12) by multiplying a smooth transition term from small $N$ region to large $N$ region as below, such that it can be applied to the whole region,

$$A_s = \tanh(2N) \cdot (10^\beta \cdot u_b^{1-1/m})^{-m}. \tag{14}$$

Combining Eq. (6), (8) and (9) with $C_{\mathrm{LW}} = 10^\beta$, we can get $C_s$ in regularised Coulomb sliding parameterization as

$$C_{\mathrm{S}} = \frac{10^\beta \cdot u_{\mathrm{b}}}{N \cdot (1 - 10^{m\beta} \cdot A_{\mathrm{S}} \cdot u_{\mathrm{b}}^{1-m})^{1/m}}. \tag{15}$$

Eq. (14) and Eq. (15) determine the conversion of the inverted friction coefficients to the regularised Coulomb sliding parameterization.

### 3.3 Ice-shelf basal melt parameterization

Sub-shelf melting of TG is influenced by ocean heat intrusions imposed by mCDW, which may be modified by several processes such as the Antarctic Slope Current and upstream coastal freshening (Nakayama et al., 2021). But we do not consider the complex ocean circulation here. Instead, we use a simplified parameterization to mimic the spatial pattern and magnitude inferred from satellites (Rignot et al., 2013; Depoorter et al., 2013; Nakayama et al., 2021) and ocean model simulations (Richter et al., 2022). We consider sub-shelf melt as a sum of low melt which plays a dominant role in the

shallow water and high melt which plays a dominant role in the deep water near the GL. We express mass balance in ice equivalent units.

We parameterise the sub-shelf melt rate in the shallow water using a linear regression of ice shelf bottom depth and ice shelf basal melting rate as modelled by the 2 km spatial resolution, tide and eddy-resolving Whole Antarctic Ocean Model

(WAOM v1.0; Richter et al., 2022) as follows:

$$M_{\mathrm{d}} = 450.284 \times \frac{1}{d + d_0} + 1.1917, \tag{16}$$

where $M_{\mathrm{d}}$ represents the basal melting rate of the ice shelf (unit: m yr$^{-1}$ i.e.), and $d$ indicates the depth of the ice shelf bottom (unit: m), $d_0 = 100$ m ensures that the denominator term is nonzero.

The WAOM v1.0 estimates of Antarctic ice shelf basal mass loss agree with satellite observations in many regions (Richter et al., 2022). However, for the TG subregion, the satellite estimates (Rignot et al., 2013; Depoorter et al., 2013; Liu et al., 2015) suggest higher basal melting (area average melt ≥10 m yr$^{-1}$) than that obtained with WAOM v1.0 (Richter et al., 2022). The area-average basal melt rate of Totten ice shelf estimated by Rignot et al. (2013) is 63.2 m yr$^{-1}$. Gwyther et al. (2018) used the Regional Ocean Modelling System (ROMS; Shchepetkin and McWilliams, 2005) that gives the area averaged basal

melting of 5 m yr$^{-1}$, and maximal melt rates exceeding 30 m yr$^{-1}$. Greene et al. (2018) also simulated the sub-shelf melt rate



distribution of the TG using ROMS, and found area averaged basal melt rates of 10-15 m yr$^{-1}$, and a maximum melt rate exceeding 80 m yr$^{-1}$ near the deep GL of the inner Totten ice shelf (TIS). Melt rate is higher in the inner than the shallow outer TIS. Nakayama et al. (2021) derived long term satellite-based estimates of ice shelf melt-rate from 1992 to 2016, which varies between 20 m yr$^{-1}$ and 80 m yr$^{-1}$. Vaňková et al. (2023) measured in-situ basal melt along the GL using
autonomous phase-sensitive radars, and found large spatial differences which could be explained by water column thickness variations.

To account for the high melt rate in the deep water, we add an extra melt rate term as

$$M_e = \begin{cases} M_{max} \times \left( \dfrac{d}{2000} \right), & d < 2000 \text{ m}; \\ M_{max}, & d \geq 2000 \text{ m}. \end{cases} \quad (17)$$

where $M_{max}$ is the maximum basal melt rate, and $d$ is the depth of the bottom surface of the ice shelf (unit: m).

In addition, we use two scaling factors $S_w$ and $S_i$ to reflect the influence of cavity geometry and avoid numerical instability caused by a vanishingly thin ice shelf, following Gladstone et al. (2017).

$$S_w = \tanh\left( e\frac{H_w}{H_{w0}} \right), \quad (18)$$

where $H_w$ is water column thickness, given by $H_w = z_i - b$ where $z_i$ is the ice shelf bottom elevation and $b$ is the bed elevation. We set $H_{w0} = 200$ m as a reference water column thickness. $S_w$ approaches 1 in deeper water ($H_w > H_{w0}$).

$S_i$ is an ice-shelf depth-scaling parameter, given by

$$S_i = \max\left[ \tanh\left( e\frac{z_{i0} - z_i}{z_s} \right), 0 \right], \quad (19)$$

where we set $z_{i0} = -80$ m as a reference ice base elevation relative to sea level (positive upwards) and $z_S = 100$ m as a
(directionless) scaling depth. The use of $S_i$ gives zero melting for $z_i > z_{i0}$.

In a combination of Eq. (16) – (19), the sub-shelf melt rate then is parameterised as

$$M = S_w \cdot S_i \left( M_d + M_e \right). \quad (20)$$

The spatial distribution of estimated sub-shelf melt rates using different values of $M_{max}$ (0, 20, 40, 80 and 160 m yr$^{-1}$) are shown in Fig. 3.





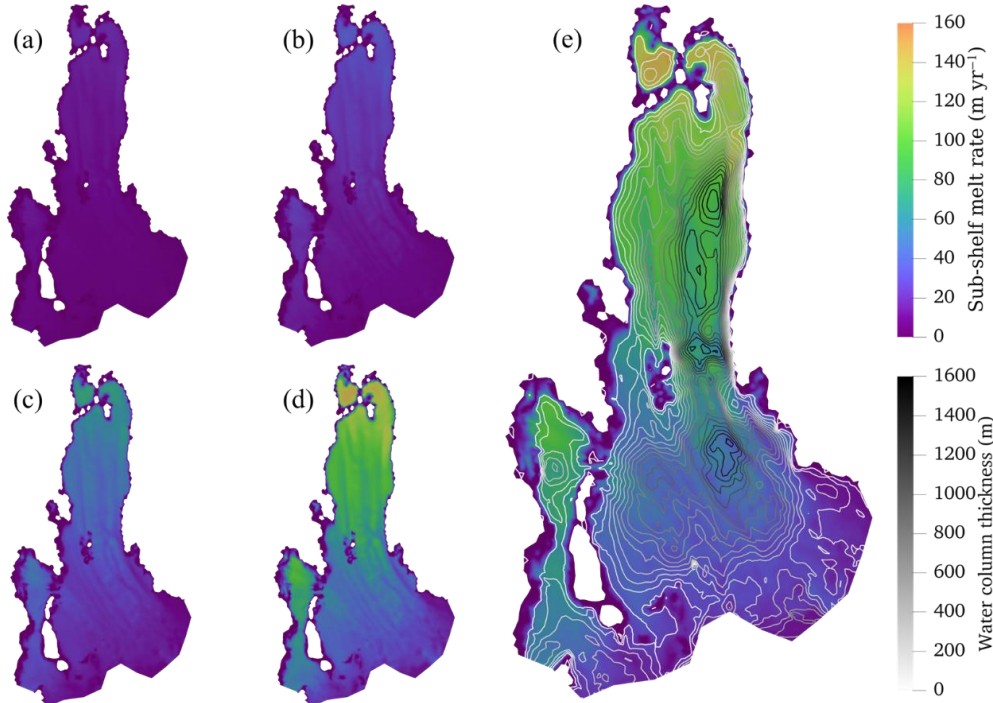


**Figure 3.** Spatial distribution of modelled sub-shelf melt rate with the maximum value of **(a)** 20 m yr⁻¹, **(b)** 40 m yr⁻¹, **(c)** 80 m yr⁻¹, **(d)** 160 m yr⁻¹. **(e)** is an enlarged version of **(d)**, with coloured contours indicating water column thickness under the ice shelf with 100 m intervals.

## 3.4 Experiment design

We firstly do a diagnostic simulation with $E_\eta = 1$, and the linear Weertman sliding parameterization, expressing the sliding coefficient by $C_{\text{LW}} = 10^\beta$ where $\beta$ is taken from the output of a whole Antarctic ice sheet inversion for the year 2015 (Gladstone et al., 2019), with a resolution in the range 4-40 km. Then we relax the free surface of the domain by a short transient run of 1 year with a small time-step size of 0.1 year to reduce the non-physical spikes in the initial surface geometry (Zhao et al., 2018; Gladstone and Wang, 2022; Wang et al., 2020).

Taking the results from surface relaxation, we use the inverse method to adjust the spatial distribution of basal friction coefficient $\beta$ to minimize the mismatch between the magnitudes of the simulated and observed (Fig. 1c; Table 1) surface velocities. We obtain the optimal spatial field of $\beta$, an updated modelled ice velocity field, and the stress field. We then perform a viscosity inversion to adjust the spatial distribution of stress enhancement factor $E_\eta$ to minimize the mismatch between the magnitudes of the simulated and observed surface velocities. We obtain the optimal spatial field of $E_\eta$, and





further updated modelled ice velocity and stress fields. The inversion for basal friction coefficient $\beta$ and the stress enhancement factor $E_\eta$, follows the same approach as in Gladstone and Wang (2022) and Wang et al. (2020). We take this state as the one for the initial year 2015.


With the modelled basal shear stress and basal sliding velocity, we convert the basal drag coefficient from the linear Weertman parameterization to those representing non-linear Weertman and regularised Coulomb parameterizations using the method described in Section 3.2. Using the three different sliding parameterizations and different sub-shelf melt rates, we carry out two sets of sensitivity tests (Table 2) and run for 35 years, from 2015 to 2050, with a time-step size of 0.1 year. A

so-called 'reference run' is chosen to illustrate more dynamic process during the 35 years simulation. We chose regularised Coulomb parameterization as the basal sliding in the reference run because it is arguably the most physically sound. Given the expectation of a warming climate, we chose for our reference run a melt rate parameterization with a maximum melt that approximates to an estimate for the temporal maximum over recent decades (i.e. 80 m yr$^{-1}$; Nakayama et al., 2021). We use maximum sub-shelf melt rates of 0, 20, 40, 80 and 160 m yr$^{-1}$ (Fig. 3) in the sensitivity tests. We use SMB modelled by

CESM2 under SSP5-8.5 scenario (Nowicki et al., 2016) for all the simulations 35 years into the future. We only consider 35 years in our simulations, because solving the Stokes equation for Totten glacier subregion is computationally expensive.

Table 2. Summary of sensitivity experiments.

| Experiment group | Experiment | Sliding parameterization | Maximum melt rate (m yr$^{-1}$) |
|---|---|---|---|
| Reference run | C_$m_{80}$ | regularised Coulomb | 80 |
| Sliding parameterization group | LW_$m_{80}$ | linear Weertman | 80 |
|  | NW_$m_{80}$ | nonlinear Weertman | 80 |
| Melt rate group | C_$m_{160}$ | regularised Coulomb | 160 |
|  | C_$m_{40}$ | regularised Coulomb | 40 |
|  | C_$m_{20}$ | regularised Coulomb | 20 |
|  | C_$m_0$ | regularised Coulomb | 0 |

## 4. Results

**4.1 Initial state and reference run result**

We obtain the inverted optimal spatial distribution of basal sliding coefficient exponent $\beta$ (Fig. 4a) and the stress enhancement factor (Figs. 4b-c) for the initial year 2015, and keep them fixed in all the prognostic simulations. $\beta$ is smaller





in the fast flow region where ice speed >10 m yr$^{-1}$, and larger in the slow ice flow region. $E_\eta$ remains at unity value over most of the grounded region; $E_\eta$ >1 in the vicinity of the GL indicates relatively stiffer ice, and $E_\eta$ <1 on the ice shelf

indicating more rapid deformation likely due to shear-band damage. For the initial year 2015, the basal shear stress (Fig. 4d) for the initial state is concentrated in the range of 0-400 kPa. The basal effective viscosity (Fig. 4e) is smaller in regions with relatively high ice temperature (Fig. 1d) and fast flow speed (>10 m yr$^{-1}$), and larger in the upstream regions with low ice temperature and slow flow speed. Shear margins of lower viscosity exist on both sides of the TG ice shelf.

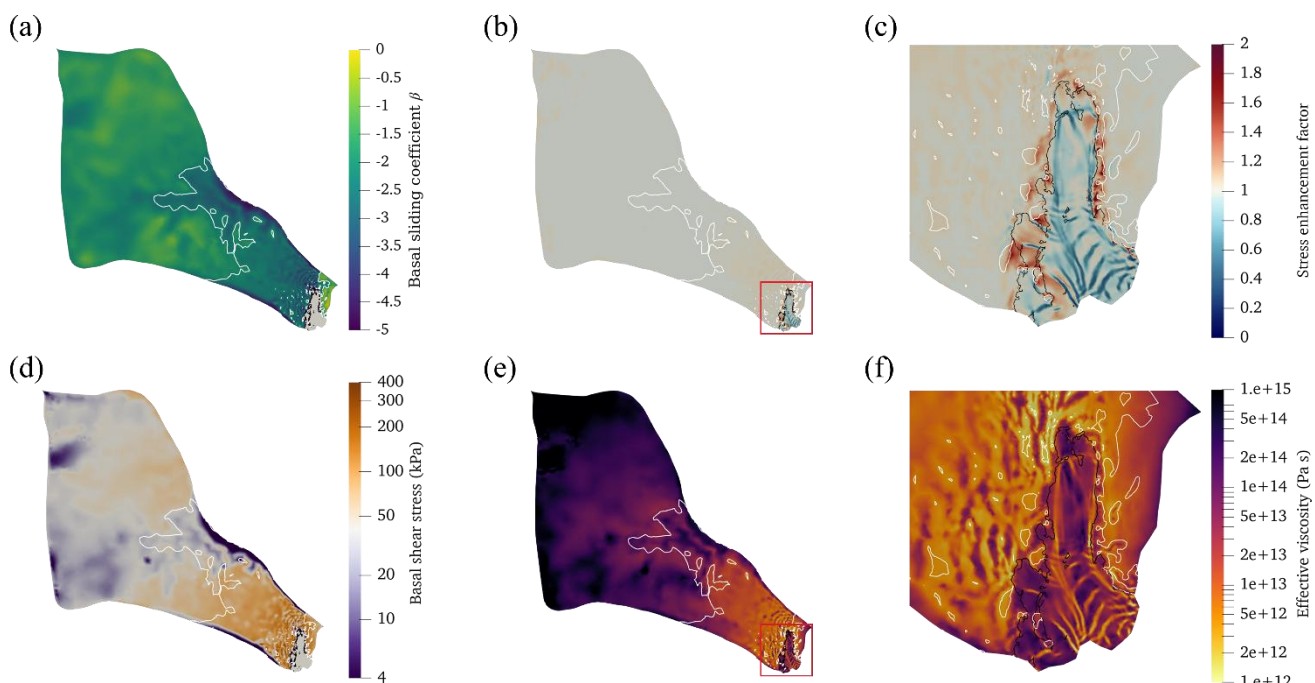

**Figure 4.** Spatial distribution of modelled **(a)** basal sliding coefficient $\beta$, **(b)** stress enhancement factor $E_\eta$, **(c)** enlarged stress enhancement factor in the red-boxed region in panel **(b)**, **(d)** basal shear stress $\tau_b$, **(e)** basal effective viscosity, **(f)** enlarged basal effective viscosity in the red box in panel (e). The solid black curves represent the GL and the solid white curves show modelled basal speed contours of 10 m yr$^{-1}$.


In the reference run, ice shelf thickness decreases significantly, and the ice shelf decelerates over the 35-year simulation. But there are large spatial differences in ice thickness change of the grounded ice. We select two flowlines for analysis: FL1 along the main trunk, and FL2 on the eastern branch (Fig. 5) of TG. The fast-flowing regime in the grounded ice becomes larger and the ice shows dynamic thinning at the eastern and southern GL (Fig. 5b, e, f). But the ice flow speed and ice

thickness are only slightly changed in the north-western GL along the flowline FL1 (Fig. 5b, c, d) and the western side of



TIS. The western side of TIS is grounded on Law Dome (Fig. 1). The GL retreats primarily along the southern and eastern grounding zones, which is consistent with previous studies (e.g. Pelle et al., 2021).

**Figure 5.** Surface velocity at the beginning of the initial year 2015 **(a)** and the end of the year 2050 **(b)** in the reference run. The solid black lines and white lines in **(a)** and **(b)** represent the GL positions in the year 2015 and 2050 respectively. Pink and purple solid lines in **(a)** and **(b)** represent flowlines FL1 and FL2 as labelled. **(c-f)** The solid color portions of the figures show the ice flow velocity (upper colorbar) profiles along FL1 **(c, d)** and FL2 **(e, f)** in the reference run in the initial year 2015 **(c, e)** and the end year 2050 **(d, f)**, with bedrock in dark grey and seawater in blue. The geometry change of TG is marked with colored solid lines (lower colorbar) for the years 2015, 2020, 2025, 2030, 2035, 2040, 2045 and 2050. The vertical elevations are exaggerated by a factor of 25. The vertical stripes in panel (c)-(f) are artifacts of plotting using Python.

## 4.2 Sensitivity experiments results



The GL at the end of the year 2050 from the sliding parameterization and melt rate groups in Table 2 are shown in Fig. 6. The GL mainly retreats in the southern and eastern grounding zones, and GL evolution is sensitive to both sliding and sub-shelf melt rates (Fig. 6). The GL is stable in the $C\_m_0$ and $C\_m_{20}$ experiments and retreats in other experiments with higher

basal melt rates (Fig. S1). The modelled GL retreats most using the linear Weertman sliding parameterization, a moderate distance using the regularised Coulomb sliding parameterization, and least using the nonlinear Weertman sliding parameterization (Fig. 6a).

The grounded ice near the GL along both flowlines is thinned more with the regularised Coulomb or linear Weertman sliding

parameterization than that using the nonlinear Weertman sliding parameterization (Fig. 7, 8). The linear Weertman sliding parameterization is derived under the simplifying assumption of Newtonian flow, which is not realistic for ice. We expect that the modelled results using the regularised Coulomb and nonlinear Weertman sliding parameterizations to be more realistic. As the ice thins, the effective pressure used in the regularised Coulomb is reduced, leading to more basal sliding and faster ice speed. Therefore, the GL retreats more using the regularised Coulomb sliding parameterization than with the

nonlinear Weertman sliding parameterization. This difference in GL retreat is large along FL2 where the bedrock slope near the GL is small, but small along FL1 where the bedrock slope near GL is large (Fig. 6, 7, 8).

Along FL1, the modelled GL with both linear Weertman sliding and regularised Coulomb sliding retreat similar distance, 8.08 km, and with nonlinear Weertman sliding it retreats 5.94 km (Fig. 7) over the period 2015-2050. Along FL2, the GL

retreats 12.09 km using linear Weertman, 10.83 km using regularised Coulomb, and only 1.32 km using nonlinear Weertman sliding parameterizations (Fig. 7). By the end of the year 2050, the grounded area loss is 3149 $km^2$, 2611 $km^2$, 1105 $km^2$ with linear Weertman, regularised Coulomb and nonlinear Weertman sliding parameterizations, respectively (Fig. 8).

In the Melt rate group of experiments (Table 2), higher basal melt rates lead to more GL retreat, thinner ice in both the ice

shelf and the grounded ice near the GL, smaller effective pressure, hence faster ice flow speeds, and larger basal drag and larger basal shear stress, which is implied by the difference between surface and basal ice speed (Fig. 7, 8). Surface and basal ice speed difference and its spatial variability (Fig. 7, 8) imply the magnitude of both horizontal shear and vertical shear and their derivatives, which would be undetectable in simplified ice dynamics models (Greve and Blatter, 2009), therefore justifies the use of full-Stokes model. Despite the differences, all the experiments show similar trajectories of GL retreat,

which is clearly controlled by basal topography (Fig. 6). With a maximum sub-shelf melt rate of 160 m $yr^{-1}$ - that is at least double the present estimates in literature - the experiment $C\_m_{160}$ leads to GL retreat at Law Dome (Fig. 6b), which is not observed in any other experiment. Ice thickness change of the ice shelf near the GL is highly dependent on basal melt rates. The ice becomes thicker using maximal basal melt rate smaller than 40 m $yr^{-1}$, and thinner when basal melt exceeds that value (Fig. S1).




The experiments $C\_m_{160}, C\_m_{80}, C\_m_{40}$ in Melt rate group (Table 2) show that the glacier GL retreats 11.29 km, 8.08 km and 5.94 km over the period 2015-2050 along FL1, while $C\_m_{20}$ and $C\_m_0$ do not yield any retreat. The experiments $C\_m_{160}, C\_m_{80}, C\_m_{40}, C\_m_{20}, C\_m_0$ result in GL retreat of 12.33 km, 10.83 km, 7.89 km, 5.14 km and 2.83 km along FL2 by the end of the year 2050, and the grounded area is reduced by about 3405 km², 2612 km², 1519 km², 916 km² and 289 km²,

respectively (Fig. 8).

The sensitivity of sub-shelf cavity thickness to basal sliding parameterization varies spatially. The three sliding parameterizations give almost the same sub-shelf cavity thickness along FL1 (Fig. 7), but the sub-shelf cavity thickness using the regularised Coulomb sliding parameterization differs from that using the other two sliding parameterizations along

FL2 (Fig. 8). The thickness of the sub-shelf cavity is sensitive to sub-shelf melt rates (Fig. 7, 8). Higher basal melt rates result in more rapid ice shelf thinning and larger sub-shelf cavities.

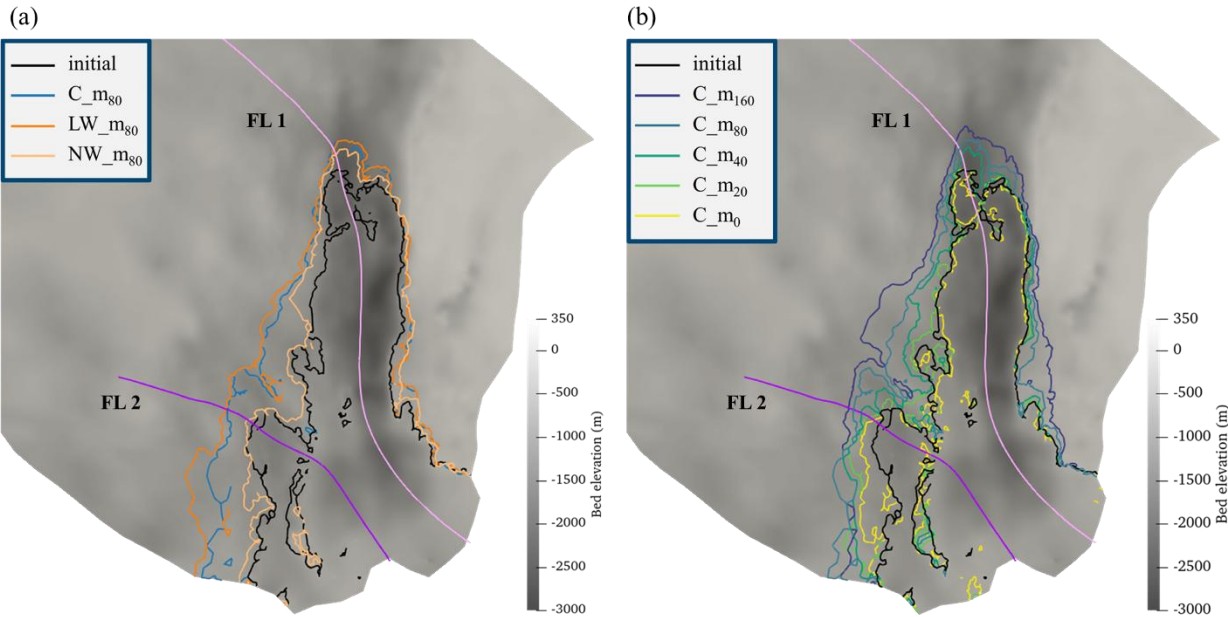

**Figure 6.** GL positions at the end of the year 2050 (coloured curves) from experiments (Table 2) in the Sliding parameterization group **(a)** and the Melt rate group **(b)**. The solid black line represents the initial GL position in the year 2015. The pink and purple solid lines are FL1

and FL2, respectively. The grey background shows the bed elevation.





**Figure 7.** Surface ice speed **(a, b)**, ice speed difference **(c, d**; surface minus basal**)**, ice thickness change **(e, f)**, and ice-sheet profiles **(g, h)** along FL1 in the initial year 2015 (black solid line) and the year 2050 (coloured solid line) from experiments (Table 2) in the Sliding parameterization group **(left column)** and Melt rate group **(right column)**. GL positions are marked with black vertical dashed line for the year 2015 and coloured vertical dashed line for the year 2050. The figure is shaded to show the geometry of the TG along the flow line in 2015, where dark grey is the bedrock, light blue the ice shelf and dark blue seawater. The elevations are exaggerated by a factor of 25.





**Figure 8.** The same as Figure 7 but for FL2.


VAF change is also sensitive to the choice in both basal sliding parameterizations and the sub-shelf melt rates. The modelled VAF loss of the TG sub-basin from the Sliding parameterization group experiments over the period 2015-2050 is equivalent to global sea level rise of 5.67 mm, 5.48 mm and 3.29 mm using linear Weertman, regularised Coulomb and non-linear Weertman sliding parameterizations, respectively (Fig. 9).


The modelled VAF loss of TG sub-basin from the Melt rate group experiments over the period 2015-2050 is equivalent to global sea level rise of 2.8 mm, 3.58 mm, 4.22 mm, 5.48 mm and 7.47 mm, using sub-shelf melt rate parameterizations with maximum values of 0, 20, 40, 80 and 160 m yr$^{-1}$ (Fig. 9d).



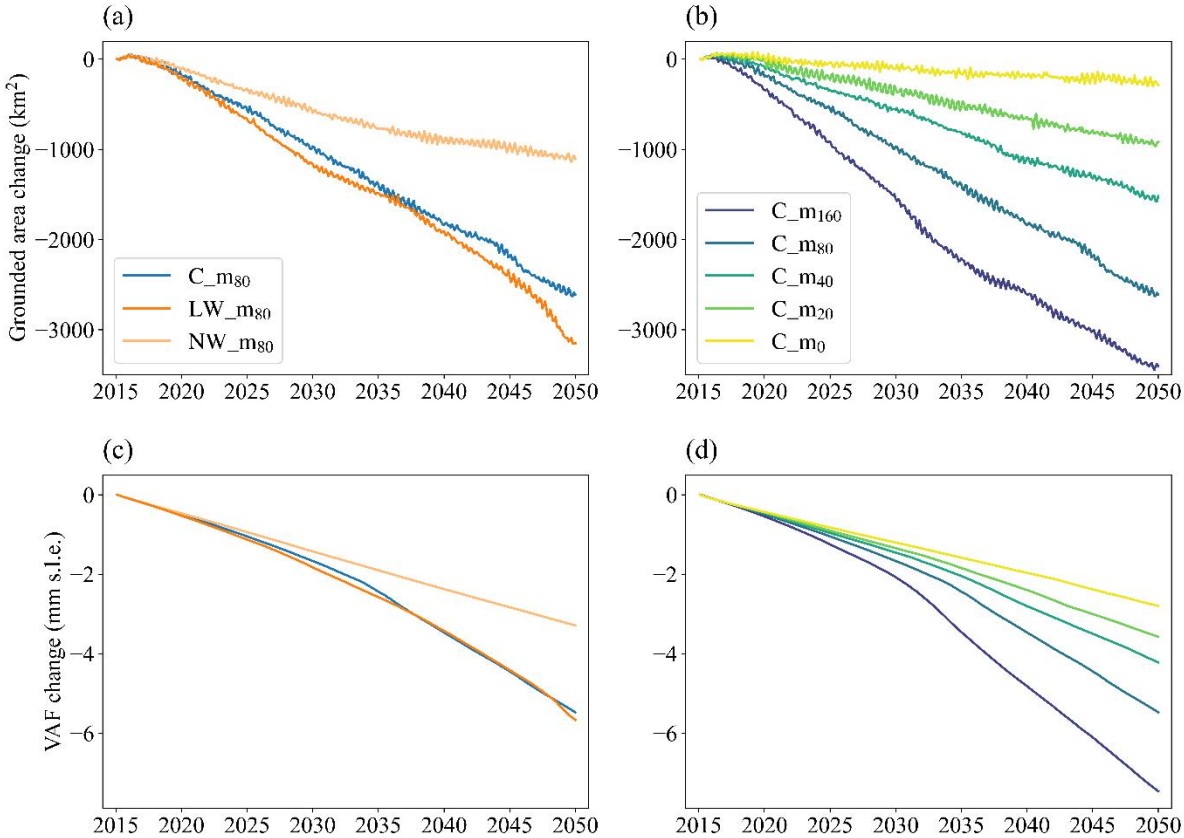

**Figure 9.** Grounded area change **(a, b)** and VAF change **(c, d)** of Sliding parameterization group **(a, c)** and Melt rate group **(b, d)** over the period 2015-2050.

## 5. Discussion

The effect of the exponent value, *m* in Eqn. (2), in the Weertman sliding parameterization has been investigated in the literature (Brondex et al., 2017, 2019; Nias et al., 2018). However, the relationship between the value of *m* and the rates of retreat and ice loss is unclear. Some simulations using the nonlinear Weertman sliding parameterization with *m*=3 respond faster to the changes in forcing (Sun et al., 2016), or lead to more loss of VAF (Brondex et al., 2019) than linear Weertman sliding parameterization. For Pine Island Glacier, Nias et al. (2018) show that *m*=3 can produce higher or lower changes in VAF depending on the bathymetry product being used. Barnes and Gudmundsson (2022) claimed that this relationship contains complexities and can be affected by regional differences, so there is no general statement to be made about what the general effect of increasing *m* will be on a system. In our study, the nonlinear Weertman sliding parameterization yields



generally less GL retreat and less ice loss than linear Weertman in the western side of TG ice shelf, and the effect of larger *m* has regional differences.

The sub-shelf melt parameterization used here is mainly controlled by the maximum melt rate near the GL. We do not consider the circulation processes such as plumes that occur in the ice shelf cavity, especially the invasion process of mCDW and changes in ocean temperature, salinity, and their variability after the expansion of the ice shelf cavity. Some studies obtain the distribution of sub-shelf melt rate by coupling ice sheet with ocean models. Pelle et al. (2021) used an asynchronously coupled ice-ocean model to project the evolution of TG to 2100. The maximum in their modelled sub-shelf melt rate is along the western TIS and exceed 80 m yr$^{-1}$ under the SSP5-8.5 scenario. They also projected significant retreat on TG's eastern and southern sectors. McCormack et al. (2021) investigate Totten's response to variable ocean forcing, using the Ice-sheet and Sea-level System Model (ISSM; Larour et al., 2012) coupled to the Potsdam Ice-shelf Cavity mOdel combined with a Plume model (PICOP; Lazeroms et al., 2018). They found the GL is steady and close to present-day only for current ocean temperatures and retreats for temperature increases ≥0.2°C mainly on the eastern and southern sectors. Both ocean temperature variability and topography control GL stability. Our modelled GL retreat occurs with relatively higher basal melt rate (≥ 40 m yr$^{-1}$), and mainly on the eastern and southern grounding zones, and is qualitatively consistent with these previous studies (Pelle et al., 2021; McCormack et al., 2021) though the magnitude and spatial pattern differ from ours.

Effective pressure has been included in several basal sliding parameterizations (e.g. Budd et al., 1984; Schoof, 2005; Tsai et al., 2015; Brondex et al., 2017, 2019) to represent the effect of water at the ice/bed interface. These studies with effective pressure-dependent basal sliding parameterizations usually adopt the assumption of perfect hydrological connectivity to the ocean (Morlighem et al., 2010; Tsai et al., 2015; Gladstone et al., 2017; Brondex et al., 2017, 2019). Reducing the effective pressure allows more water to flow through the basal network and enhances basal slip (Zwally et al., 2002; Stearns et al., 2008). Simulations using Weertman sliding parameterization produce less VAF loss than effective pressure-dependent laws like Budd and regularised Coulomb sliding parameterization assuming a perfect hydrological connection between the subglacial drainage system and the ocean (Brondex et al., 2017, 2019). Therefore, the choice of effective pressure in basal sliding parameterizations has significant effect on the modelled ice dynamic evolution. In this study we take the effective pressure as 5% of overburden pressure from subglacial hydrology modelling (Dow et al., 2020). The regularised Coulomb sliding parameterization yields more VAF loss than nonlinear Weertman sliding parameterization. This is consistent with previous studies (Brondex et al., 2017, 2019), despite our effective pressure values taken from a subglacial hydrology modelling is less responsive to ice thickness changes than the widely used assumption of perfect hydrological connection to the ocean.



## 6. Conclusions

We find that our simulated GL retreat and VAF loss of Totten glacier using regularised Coulomb sliding parameterization is
more than that using nonlinear Weertman sliding parameterization. This agrees with previous findings in the literature that effective pressure-dependent sliding parameterizations usually produce more VAF loss than Weertman-type sliding parameterizations. This is despite the fact that our effective pressure parameterization is only 5% as responsive to ice thickness changes as the widely used assumption of perfect hydrological connection to the ocean. Our simulated GL retreat and VAF loss using linear Weertman sliding parameterization is larger than that using nonlinear Weertman sliding
parameterization. There are no general statements to be made about the effect of different exponents used in Weertman sliding parameterizations, because regional bathymetry and basal velocity play large roles as well as sliding law.

The modelled GL retreats when maximal basal melt rate is $\geqslant$ 40 m yr$^{-1}$, and mainly along the eastern and southern grounding zones. This is consistent with previous studies, although with large differences in local retreat pattern and
magnitude. The differences in pattern are mainly due to the differences in ocean thermal forcing and hence the details of basal melt rate parameterizations and model setup. More long-term ocean state observations would reduce the uncertainty in future projections by allowing both spatial and temporally realistic variability in sub ice shelf met rates. It seems that TG does not face the risk of marine ice sheet instability in the near future because of the prograde slope in the first tens of kilometres inland of the GL. Our simulations do not retreat as past this slope before the year 2050. Once the GL retreats
further onto the retrograde bed slope then we would expect far faster GL retreats in the long term. The change of sub-shelf cavity thickness is not sensitive to the choice of basal sliding parameterization; however, it is sensitive to sub-shelf melt rates. This indicates the need to consider geometry change of sub-shelf cavity and water circulation inside the sub-shelf cavity when using ocean model or plume model for long term projections.

*Data availability.* MEaSUREs Antarctic Boundaries for IPY 2007–2009 from Satellite Radar, version 2, is available at https://doi.org/10.5067/AXE4121732AD (Mouginot et al., 2017). MEaSUREs BedMachine Antarctica, version 2, is available at https://doi.org/10.5067/E1QL9HFQ7A8M (Morlighem, 2020). MEaSUREs InSAR-based Antarctic ice velocity Map, version 2, is available at https://doi.org/10.5067/D7GK8F5J8M8R (Rignot et al., 2017). Ice temperature of SICOPOLIS output (Greve et al., 2020) is available upon request to the author. SMB of CESM2 under the SSP5-8.5 scenario
is available from the Globus ISMIP6_Forcing endpoint, at https://app.globus.org/file-manager?origin_id=ad1a6ed8-4de0-4490-93a9-8258931766c7&origin_path=%2F (Nowicki et al., 2016). Elmer/Ice is a free and open-source software, published on Zenodo (Elmer/Ice Authors, 2022, https://doi.org/10.5281/zenodo.7892181).

*Competing interests.* The contact author has declared that none of the authors has any competing interests.





*Acknowledgments.* This work was supported by National Key Research and Development Program of China (grant no. 2021YFB3900105), and Academy of Finland (grant no. 322430 and 355572), the Finnish Ministry of Education and Culture and CSC-IT Center for Science (Decision diary number OKM/10/524/2022).

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
