# Peer review of "Sensitivity of Totten Glacier dynamics to sliding parameterizations and ice shelf basal melt rates"

_EGUsphere, 2024_

## Referee Comment (RC1)

**Review Sensitivity of Totten Glacier Dynamcis to Sliding Parameterizations and ice shelf basal melt rates.**

In this study, the authors assessed the effect of applying three different basal sliding parameterizations (linear Weertman, non-linear Weertman and regularized Coulomb sliding) and different sub shelf basal melt rates on the modelled evolution (future retreat) of Totten Glacier, East Antarctica. This is an interesting paper, applying a sensitivity analysis on a less studied but nevertheless important area of the Antarctic Ice Sheet, with some relevant results. I enjoyed reading it! It general it is well written, but lacks quantification to back up some key claims. Furthermore, a better justification/explanation on the inversion procedure and for the choice of Full Stokes would be beneficial, see the two major points below.

Major points:
- The authors apply the Full Stokes approximation and argue that this is necessary or justified by considering the spatially varying velocity fields or the velocity gradients. I like and support the use of a Full Stokes solver because, as the authors mention in this manuscript, all deviatoric stresses are resolved. This is especially relevant at small scales and at or close to the grounding line. Only, a major disadvantage is the runtime: the authors mentioned to be limited to 35 model years. To me this is very short, when assessing sensitivities to parameterizations you would like to see longer runtime/more retreat. For this study, I would prefer to see longer simulations with a simpler approximation (DIVA, BP, SSA+SIA), also considering the idealized forcing. If this is outside of the scope/computational expenses, I would at least like to see a comparison with a single iteration with a simpler approximation to quantify the added value of the Full Stokes solver (in terms of for example stresses at the grounding line, ice surface velocity error). Another suggestion could be to use realistic oceanic forcing (from the ISMIP6 project for example), because to me the closer-to-reality FS solver is partly counteracted by a further-from-reality melt parameterization.
- I am confused by your inversion method. You both metion Gladstone et al 2019 and Gladstone and Wang 2022 as sources for your inversion. The former links to an empty DOI, the latter to a general description on inversion in Elmer Ice applied to Pine Island Glacier. I would like to see more precisely what you did and what equations where used. For example: did you use inverted beta's from Gladstone et al 2019? How did you interpolate them to your grid? Are the runs similar enough that you can 'copy' an inverted field from one simulation to the other? What equations did you use from Gladstone and Wang 2022? See more detailed comments below.

Line by line specific points:
- Ln 35: a discussion on a marine ice sheet instability like retreat could be added here. Is TG susceptible to MISI?
- Ln 37: where on the continental shelf? All around the AIS? Or only close to TG? I was in the understanding that it currently is mainly present in the Amundsen Sea Embayement.

- Ln 47: How much is 'significant retreat'? Under which climate warming scenarios?
- Ln 60: 'A nonlinear Weertman...' I do not understand this sentence, could you split it up or rephrase?
- Ln 74: what could be added here is the notion that Weertman sliding was originally developed for slow ice on hard beds (East Antarctica) and coulomb sliding more for faster outlet glaciers such as Pine Island and Thwaites. I believe this is featured in the ABUMIP paper by Sainan Sun.
- I couldn't make sense of some of the equations in the method section due to what I expect is a PDF rendering error: some symbols appear as questionmarks. I am not sure where the error lies, but this made it impossible for me to assess these. It happened in Ln 142 – Ln 155
- Ln 81: under which warming scenarios?
- Ln 91: 'Has more complete physics' is a very broad statement and one that does not bring enough credit to other modelers groups in the world. I would rephrase this to something like 'Elmer/Ice has the option to run with the full stokes deviatoric stress tensor, which is an unique feature of this model'.
- Ln 91: what are 'the simplified models'?
- Fig 1: consider adding ice thickness as well, either as contour lines in the first plots or as a separate panel. Also, to remove the amount of numbers showed in these plots you can remove some of the axis labels. For example, in the top row, figure a,b and c share the same y-axis. Just showing it for figure a would be enough I think.
- Fig 1c: I would suggest to use a logarithmic scale for this plot.
- Fig 1e: Which reference did you use for these drainage basins?
- Fig 1: the purple GL is almost invisible. Also, I would suggest to show the GL for the whole area, not just for Totten.
- Fig 2: It is unclear to me which subfigure belongs where. The main blue discretization should be labeled (a), right? What is the use of showing the upper left red outline? Why are the observed velocities (are those ice surface velocities as well?) so much smaller than in Figure 1c? Also, you're scale bar looks off to me. Subpanel c should be about 400 km in width and maybe 500/600 km in lenght according to its own scalebar, while the red square in subpanel b is about 200 km in dimensions according to the bigger scale bar. I would also suggest to use a different color for the discretized elements in subfigure b. blue is already used in the two colormaps showing ice thickness and ice surface velocities.
- Ln 137: which inverse method? And what target variable did you use?
- Ln 159: You could add a discussion on the not represented GIA. For 35 years it likely does not matter, but it would still be nice to read that and why.
- Ln 170: You could make Eq 6 and Eq 7 one equation, and just use m=1 for the linear cases. This saves space and does not require you to write $C_{lw}$.
- Ln 172: In Eq 8 the vector **u_b** is used twice. I think the first one should be the magnitude $u_b$.
- Ln 181: You could add Leguy et al 2014 and Leguy et al 2021, where they used a parameterization based on the height above floatation to mimic some hydrological connection for ice resting on bedrock below sea level.
- Ln 190: In what quantifiable way does this alter your simulations?

- Ln 191: What is a linear Weertman sliding inversion? Are there different inversions per basal friction law?
- Ln 196 -212: Am I understanding correctly here that, in the case of rewriting Weertman to Coulomb sliding, you could not find an exact solution for the free parameter in the coulomb sliding law and you had to revert to limits to find $C_s$ with the inclusion of a smoothing term in Eq 14? If that is true, then, I like the elegance of this method but I am then not convinced that plugging the $C_s$ you found in Eq 15 in Eq 8 will give you the same $\tau_b$ as Eq 7. Is that correct? And if so, can you then show that the deviation of $\tau_b$ is small and that it has no to little impact on a continuation simulation?
- Ln 220: what is shallow and what is deep water?
- Ln 225: Eq 16: where did you get the (non linear!) regression, and how well does it do in representing the values from the WAOM? A scatter plot between the WAOM basal melt rates versus draft depths, with this linear regression fitted through would make this clearer.
- Ln 225: Eq 16: I am also not sure about the addition of $d_0$ here. In my view, d is always nonzero if there is ice present in a grid element. If there is no ice present, one does not need to calculate the melt rates. Adding 100 is quite substantial, say your draft depth is also 100 m (it typically is between 0 and 1000 m, order of magnitude), adding 100 to your denominator for the sake of preventing zeros will alter your results significantly. Can you not remove this $d_0$?
- Ln 245 – 255: I am missing a discussion of the physical interpretation of the $S_i$ and $S_w$ values, just stating that the reflect the influence of cavity geometry and avoid numerical instability is in my opinion not enough. How does the cavity geometry influence the melt rates, and how is that reflected by $S_w$? Same for the numerical instability.
- Fig 3: consider adding the observations to this figure as well.
- What is inverted for is clear to me, the basal friction parameters C. However, how this is done (nudging, data assimilation) and with what as target (ice thickness, velocity) is not clear to me. Are you using the inversion results of Gladstone et al 2019? The reference here runs to an empty DOI, so I could not check what inversion you are using. Immediately after that you state the resolution of that simulation to be 4-40 km. Your simulation uses up to 900 meter resolution, how are you interpolating the inverted values without introducing model drift? What's more (and possibly more important) you are using the newest Bedmachine dataset of Morlighem, 2020, Gladstone et al 2019 could not have used this, so they inverted a friction parameter using a different bedrockheight dataset. Also, your temperature profile from SICOPOLIS is different. You have to convince me now that you can take inverted fields from a different model run with different input datasets, approximations and parameterizations, and without problems use it in your own setup. You mention another study at the end of this paragraph (Gladstone and Wang, 2022) where some explanation is given but for Pine Island instead of Totten. I would suggest to add the equations you use from this paper and copy them to your study.

- Ln 266: A diagnostic simulation tells me that you already did some kind of spinup or initialization. I would mention here your spinup procedure, and mention the inversion procedure as well.
- Ln 272: 'the inverse method'. What inverse method? Please specify.
- Ln 275: If I get it correctly, you target surface ice velocities in your inversion, first with basal friction and then with the viscosity. It would be nice to read something here on this serial approach, why not in parallel? And what was the effect of this extra inversion step with the flow enhancement factor? Can the basal friction inversion alone not give the right ice surface velocities? Also, what was done by Gladstone et al 2019?
- Ln 286: Why is regularized coulomb the most physically sound? Provide references.
- Ln 290: SMB has already been mentioned, consider removing it here.
- Ln 291: I would argue that, if computational expenses are too high when running Full Stokes, shift to a faster approximation (Hydrostatic, Blatter-Pattyn or for example DIVA) and run further into the future. 35 years is short to make statements about sensitivities to basal friction, maybe the simulations will start to deviate as soon as the grounding line retreats further (e.g. after 100-200 years). Or converge to some steady state upstream. Doing 35 year simulations in my opinion is particularly usefull when making state-of-the art projections of glacier retreat, with forcing from CMIP models. For sensitivity studies like this one, I would recommend to run longer.
- Ln 296: you do not keep beta fixed right? You rewrite them to fit with other friction parameterizations. Please state so. Also, please make clear which beta (the one from Gladstone et al 2019 or one obtained after your own relaxation) was used.
- Fig 4: the grounding line is very hard to see, consider changing the colors and/or the thickness. Also I would like to see the observed grounding line position next to the modelled one to asses how well your model performs.
- Ln 311: Why does the ice shelf decelerate? I would expect some speedup due to the loss of buttressing due to the loss of ice shelf thickness.
- Fig 5: a difference plot would be more informative here, since the visual difference between beginning and end of the simulation is hard to see.
- Fig 5 c-f: I appreciate the honesty when saying that the vertical lines are Python artefacts, but I would still like them to be removed before publication.
- Ln 327-333: this conclusion, that there are various grounding line retreats for different sliding laws, is not what I got from reading your abstract in which you mentioned that the basal sliding law did not matter.
- Ln 339: this is not neccesarely the case: less sliding and particular coulomb sliding will make it easier for ice to flow from far upstream to the GL, preventing the thinning at the grounding line.
- Ln 353 – 355: I do not agree here. First, the magnitudes of the spatial velocity differences might imply something on horizontal shear and its derivatives, but why is that relevant? Also, if you want to show the spatial variability in the shear stresses, why do you not plot the shear stresses themselves? But the most important point: there are multiple other approximations that take either horizontal or vertical derivatives of the shear stresses into account, or combinations of them. You can pick Hybrid SIA+SSA for example, or the Depth

Integrated Viscosity Approximation (DIVA), or Blatter-Pattyn, or the Hydrostatic Approximation. Those will all resolve the quantities you want to detect, with less computational expenses. This does not justify the need for a Full-Stokes model, and if you can only run for 35 years with Full Stokes, I would strongly suggest to run longer with a less computational heavy approximation, or at the very least rephrase and rethink why you chose Full Stokes in the first place.

- Ln 355: Despite what differences? Also, you just argued that there is a huge difference in grounding line response (10 km vs 1 km), now you are writing the opposite. From figure 6, I conclude that there is quite a difference in GL retreat when using a different sliding law, contrasting your abstract.
- Ln 355: Why is this clearly controlled by the topography? I cannot see this in Figure 6.
- Ln 370: this conclusion seems to be a bit obvious, that melt rates directly influence the cavity thickness. Whats more interesting is the relation between sliding law and cavity thickness. Can you quantify this effect? Why does another friction parameterization lead to different ice shelf cavity thickness?
- Fig 6 and fig 7: I see much more grounding line retreat difference in Fig 6 compared to Fig 7, why is that?
- Fig 9: is there a seasonal cycle in your simulations? It shouldn't be because of the simplified melt parameterization.
- Ln 396: I am missing a discussion on your inversion procedure: how did taking the fields from Gladstone et al 2019 influence your results? Was there any model drift or how did you remove it?
- Ln 433: 5% is extremely low in my opinion, I would suggest to take a look at the parameterization proposed by Leguy et al 2014 (Parameterization of basal friction near grounding lines in a one-dimensional ice sheet model)
- Overall: the short timescales are not discussed in this section, and its possible effect on the simulations. Barnes and Gudmundsson 2022 and Brondex et al 2017 and Brondex et al 2019 all conducted longer simulations, so you might find similar or non-simular results if you extend your simulations to match their lengths, typically 100-300 years.
- Ln 445: I would not ask you for general statements, but what I would like to read is your thoughts on the fact that shapewise (in a basal friction versus basal velocity plot) the non-linear Weertman and coulomb sliding law look more like each other, than the coulomb sliding law and the linear Weertman. Why are then the modelled ice sheet responses so similar between the linear Weertman and Coulomb sliding law?
- Ln 447: 'the maximal basal melt rates are'.
- Ln 449: this seems contradictory: is it consistent or are there large differences?
- Ln 452: 'Melt' has a typo
- Ln 455 - 456: Earlier on I read this: 'The sensitivity of sub-shelf cavity thickness to basal sliding parameterization varies spatially', which seems to contradict this statement. Which is true?
- Ln 465: consider publishing your scripts to make the figures and datasets of your simulations as well.

---

## Referee Comment (RC2)

**Manuscript Review**

"Sensitivity of Totten Glacier dynamics to sliding parameterizations and ice shelf basal melt rates"

Yiliang Ma, Liyun Zhao, Rupert Gladstone, Thomas Zwinger, Michael Wolovick and John C. Moore

**Summary:**

Ma et al. present an analysis of the effects of three different basal sliding parameterisations and four different ice-shelf basal melt parameterisations on the evolution of Totten Glacier. To do this they use the full-Stokes model capabilities of Elmer/Ice and perform simulations over a 35-year period from 2015-2050. They find that grounding line retreat occurs when the maximum value in the basal melt parameterisation is greater than 40 m/a. They also find that the linear Weertman sliding law generates the most grounding line retreat, closely followed by the regularised Coulomb sliding law, with the non-linear Weertman sliding law producing the least grounding line retreat and mass loss over the 35-year simulation period.

The paper is well-written, and the results provide additional insights into the effects of commonly used basal sliding parameterisations on the dynamics of an important glacier in East Antarctica when simulated with a full-Stokes model. However, I have a major point that I would like to see addressed before publication, relating to the presentation and discussion of the full-Stokes model results.

**Major Comment:**

My concern is with the presentation and analysis of the modelled ice velocities in Figures 5, 7 and 8, and in the text in lines 349-354.

Panels (c) and (d) of Figure 7 (and 8 to some extent) show large oscillations in the difference between the surface and basal ice speed along a flowline upstream of the grounding line. In some experiments, this difference approaches 800 m/a. This difference must be due to vertical shearing in the ice, and this result implies that over three-quarters of the total flow comes from vertical shearing in these fast-flowing ice stream regions. The fact that the difference between surface and basal ice speed then drops close to zero a few kilometres upstream/downstream also suggests that this large vertical shearing quickly disappears as a factor in the dynamics, and the flow is then dominated by basal slip without any clear variation in the surface or basal topography.

These estimates for vertical shearing in the ice are very large (typical values would be < 100 m/a) and don't appear to be physically plausible. It is also unexpected that there could be such profound changes in the dominant mechanism for flow over a few kilometres along a flowline without any appearance of this in the overall surface speed (as shown by the much smoother curves in panels (a) and (b) of Figure 7 when compared to panels (c) and (d)).

Could there have been an issue with the post-processing of the model data? The artefacts in Figure 5 – that the authors attribute to plotting in Python – also lead me to this as a possibility. Whilst it is hard to tell definitively, it appears that the oscillations in basal and surface speed that are clear in Figures 7 and 8 are also visible in the ice velocities plotted in panels (c) and (d) of Figure 5 and are attributed to plotting artefacts there. The pronounced gradients in the surface and basal ice speeds (in both the vertical and horizontal dimensions) shown by the stripes of different shades of green appear to be in the same locations as some of the largest oscillations in Figure 7 panels (c) and (d).

Could the authors please verify that these results are not due to an error in the post-processing of the model data? Perhaps visualising the data in ParaView and comparing it with their Python-generated plots might reveal potential discrepancies. A spatial map of the basal ice velocity would also be useful in understanding what is going on in the model output.

If this is indeed the behaviour of the full-Stokes model in this region, then I think that the authors need to expand much more on these results. This would also require a physical explanation for the readers to understand what mechanisms could be driving such large rates of vertical shearing in the ice and the large variations in its contribution to the flow over just a few kilometres in the horizontal dimension, without any expression in the overall surface ice speed.

**Minor Comments:**

Line 92: Can you explain why you expect to gain more information on basal processes from the full-Stokes model compared to the range of approximations that are available and often used? It would be good to have more justification for the benefits of your use of full-Stokes given the fact that it limits your experiments to just 35 years.

Figure 1: Could you show these plots zoomed-in on the area of interest (as in Figure 2 (c) and elsewhere)? As you have a 35-year experiment and only see limited grounding line retreat, much of the model domain is not of interest to your results, and by zooming out the reader loses much of the detail in bed elevation or flow speed that is important.

Figure 2: It's not clear what is gained by showing the inset in panel (b) here, I would consider removing it.

Line 190: I would be interested to know what impact this choice has on your results, either discussed here or in the discussion section. It seems important given your use of a pressure-dependent sliding law and the impacts you hint at here.

Line 229: Why do you need to have $d_0 = 100$ in Equation 16? d should always be > 0 on an ice shelf. Even if, for numerical reasons, it can become 0, why use $d_0 = 100$ to correct for that? Did you use (d – 100) as the value for the ice-shelf bottom depth in your linear regression to account for this constant?

Line 279: You state that this initial state is representative of 2015, but the data sets used are mosaics whose data collection period spans decades (e.g. your Table 1 shows that the ice velocity is a mosaic of data from 1996-2016). I'm not sure that it is possible to state that your initial state is 2015 without using datasets timestamped to that year – especially for a region which has seen significant changes as you outline in your introduction.

Line 291: The short timescale of 35 years makes it more important to state the benefits of using full-Stokes for these experiments to balance this limitation (see earlier point).

Figure 4: Again, I would prefer to see plots zoomed-in on the region of interest, as in panels (c) and (f) so that the details of the basal sliding and basal shear stress can be seen. The colour scale for the stress enhancement factor colour bar (white around 0) seems different to the one in the maps (grey around 0). Finally, I am not sure of the benefit of plotting the 10 m/a basal speed contours and suggest removing them.

Figure 5: See my main comments here for possible data issues, but if these are genuine artefacts of plotting in Python then they need to be corrected in updated plots.

Line 455: In the Results section (Line 366) you stated that the sub-shelf cavity thickness *did* depend on the basal sliding relation along FL2. This was surprising to me, and I would be interested to know what the physical mechanism linking the upstream conditions at the bed to the thickness of the ice-shelf cavity could be, and also the strength of this relationship compared to the much more direct impact of different basal melt rates under the ice shelf. Please clarify this discrepancy here.

---

## Author Comment (AC1)

Referee's comments are in blue, our reply in black, quotes in the revised manuscript in red.

**Review Sensitivity of Totten Glacier Dynamcis to Sliding Parameterizations and ice shelf basal melt rates.**

In this study, the authors assessed the effect of applying three different basal sliding parameterizations (linear Weertman, non-linear Weertman and regularized Coulomb sliding) and different sub shelf basal melt rates on the modelled evolution (future retreat) of Totten Glacier, East Antarctica. This is an interesting paper, applying a sensitivity analysis on a less studied but nevertheless important area of the Antarctic Ice Sheet, with some relevant results. I enjoyed reading it! It general it is well written, but lacks quantification to back up some key claims. Furthermore, a better justification/explanation on the inversion procedure and for the choice of Full Stokes would be beneficial, see the two major points below.

Major points:
   - The authors apply the Full Stokes approximation and argue that this is necessary or justified by considering the spatially varying velocity fields or the velocity gradients. I like and support the use of a Full Stokes solver because, as the authors mention in this manuscript, all deviatoric stresses are resolved. This is especially relevant at small scales and at or close to the grounding line. Only, a major disadvantage is the runtime: the authors mentioned to be limited to 35 model years. To me this is very short, when assessing sensitivities to parameterizations you would like to see longer runtime/more retreat. For this study, I would prefer to see longer simulations with a simpler approximation (DIVA, BP, SSA+SIA), also considering the idealized forcing. If this is outside of the scope/computational expenses, I would at least like to see a comparison with a single iteration with a simpler approximation to quantify the added value of the Full Stokes solver (in terms of for example stresses at the grounding line, ice surface velocity error). Another suggestion could be to use realistic oceanic forcing (from the ISMIP6 project for example), because to me the closer-to-reality FS solver is partly counteracted by a further-from-reality melt parameterization.

Reply: (1) It is expensive to run all the experiments with a full-Stokes model. So we extended the simulations using the full-Stokes model to the year 2100 with three different sliding parameterizations but the same sub-shelf basal melt rate.
(2) We also compared a single iteration in the prognostic run between a simpler approximation, B-P model, and Full-Stokes model, restarting from the same steady state. Switching from Stokes to BP halves the speed in the shelf, hence full Stokes is required in this context to achieve acceptable accuracy. We add one sentence in the discussion "Switching from Stokes to Blatter-Pattyn for a single iteration in the

prognostic run halves the speed in the shelf, which suggests the necessity of using the full-Stokes model."

(3) On your suggestion of using ISMIP6 ocean forcing, we found a presentation in EGU2020 (Haubner et al., Changes on Totten glacier dependent on oceanic forcing based on ISMIP6, EGU2020-9208) which investigated effects of ISMIP6 scenarios on the Aurora Basin. They used the ISMIP6 sub-shelf (non-local) basal melt parameterization driven by different CMIP models. They found different trends (mass loss or mass gain) with different choices of ocean model, hence there is a very large across-model spread in the ISMIP6 ocean forcing, so although some realizations may be close to reality, their distribution is wide. To help verify our melt rates we include an additional recent satellite-based estimate (Adusumilli et al., 2020) in our revision. "Adusumilli et al. (2020) estimated satellite-derived high-resolution time-averaged (2010-2018) basal melt rate map for most ice shelves in Antarctic, and found the basal melt rate of Totten ice shelf is generally 20-40 m yr$^{-1}$ with a maximum of 80 m yr$^{-1}$ near the grounding line (Fig. 3d)."

We plot the satellite-based estimates of sub-shelf melt rate by Adusumilli et al. (2020) in the updated Fig. 3(d). We find its spatial pattern and magnitude is close to our sub-shelf melt parameterization with the maximum value of 40 m/yr (Fig. 3b). This adds more confidence in our sub-shelf melt parameterization, and we do not think oceanic forcing from ISMIP6 are significantly closer to reality than our sub-shelf melt parameterization.

[Figure]

**Figure 3.** Spatial distribution of modelled sub-shelf melt rate with the maximum value of **(a)** 20 m yr$^{-1}$, **(b)** 40 m yr$^{-1}$, **(c)** 80 m yr$^{-1}$, **(e)** 160 m yr$^{-1}$ with coloured contours indicating water column thickness under the ice shelf with 100 m intervals. Plot **(d)** shows the satellite-based estimates (Adusumilli et al., 2020).

- I am confused by your inversion method. You both mention Gladstone et al

2019 and Gladstone and Wang 2022 as sources for your inversion. The former links to an empty DOI, the latter to a general description on inversion in Elmer Ice applied to Pine Island Glacier. I would like to see more precisely what you did and what equations where used. For example: did you use inverted beta's from Gladstone et al 2019? How did you interpolate them to your grid? Are the runs similar enough that you can 'copy' an inverted field from one simulation to the other? What equations did you use from Gladstone and Wang 2022? See more detailed comments below.

Reply: The doi to this paper (Gladstone, R., Zhao, C., and Zwinger, T.: ISMIP6 Projections-Antarctica read me file, Zenodo, https://doi.org/10.5281/zenodo.3484635, 2019.) is not an empty doi. Co-authors in Europe can open it. But authors based in China cannot open it. So, the problem may be caused by internet limitations. Anyhow, we attached this paper as supplement with the revision, so you can download it.

To answer "I would like to see more precisely what you did and what equations where used. For example: did you use inverted beta's from Gladstone et al 2019? How did you interpolate them to your grid? "
We rewrite this section in the revision
"The following simulations are carried out in series as part of the initialization procedure. We firstly do a steady state simulation with $E_\eta=1$ , and the linear Weertman sliding parameterization, expressing the sliding coefficient by $C_{\mathrm{LW}} = 10^\beta$ where $\beta$ is from the output of a whole Antarctic ice sheet inversion for the year 2015 (Gladstone et al., 2019), and linearly interpolated from their mesh to the finer mesh in this study. Then we relax the free surface of the domain by a short transient run of 1 year with a small time-step size of 0.1 year to reduce the non-physical spikes in the initial surface geometry (Zhao et al., 2018; Gladstone and Wang, 2022; Wang et al., 2020). Taking the results from surface relaxation, we use the variational inverse method (Morlighem et al., 2010) to adjust the spatial distribution of basal friction coefficient $\beta$ to minimize the mismatch between the magnitudes of the simulated and observed (Fig. 1c; Table 1) surface velocities. We obtain the optimal spatial field of $\beta$, an updated modelled ice velocity field, and the stress field.".

To answer "Are the runs similar enough that you can 'copy' an inverted field from one simulation to the other? " We do not 'copy' the inverted beta field of Gladstone et al. (2019) to our simulation. We only take it as an initial estimate and we interpolated it from their mesh to our finer mesh.

To answer "What equations did you use from Gladstone and Wang 2022?", the inversion for basal friction coefficient and the stress enhancement factor follows the

same approach as in Gladstone and Wang (2022). The inversions make use of Tikhonov regularization (Gillet-Chaulet et al., 2012). Two separate regularization parameters, $\lambda_\beta$ and $\lambda_\eta$, are used for the drag inversions and viscosity enhancement factor inversions. The total cost function, $J_{tot}$, is the sum of misfit, $J_0$, and weighted regularization term for the drag inversion:

$$J_{tot} = J_0 + \lambda_\beta J_{reg}.$$

$\lambda_\beta$ would be replaced by $\lambda_\eta$ for the viscosity enhancement factor inversion.

**Line by line specific points:**
- Ln 35: a discussion on a marine ice sheet instability like retreat could be added here. Is TG susceptible to MISI?
Reply: TG is not susceptible to MISI at present. But it is potential threat. We change the revision to "The mass imbalance of TG and the existence of deepening topography extending far inland have raised concerns that this subbasin has the threat of significant future grounded ice loss caused by marine ice sheet instability if the grounding line were to continue retreating (Li et al., 2016; Morlighem et al., 2020). ".

- Ln 37: where on the continental shelf? All around the AIS? Or only close to TG? I was in the understanding that it currently is mainly present in the Amundsen Sea Embayement.
Reply: Yes you are right, so we remove this statement.

- Ln 47: How much is 'significant retreat'? Under which climate warming scenarios?
Reply: For the eastern grounding zone, Sun et al. (2016) found retreat of 10 km over 200 yrs driven by HadCM3 under A1B. Pelle et al. (2021) found retreat of about 10 km by 2100 under SSP585. We add this information in the revision. "These studies project that TG will experience continuous and significant retreat on its eastern sectors by 10 km at 2200 under the A1B scenario (Sun et al., 2016) and 10 km by 2100 under the SSP585 scenario (Pelle et al., 2021)."

- Ln 60: 'A nonlinear Weertman…' I do not understand this sentence, could you split it up or rephrase?
Reply: We change this sentence "A nonlinear Weertman sliding parametrization with an exponent of 1/3 applies in the lack of large enough obstacle sizes to induce regelation processes, in which case sliding is dominated by ice-deformation on the largest occurring roughness features (Fowler, 1981)." to "If the obstacle size is not large enough to induce regelation processes, sliding is dominated by ice-deformation on the largest occurring roughness features, and a nonlinear Weertman sliding parametrization with an exponent of 1/3 applies (Fowler, 1981)".

- Ln 74: what could be added here is the notion that Weertman sliding was originally developed for slow ice on hard beds (East Antarctica) and coulom sliding more for faster outlet glaciers such as Pine Island and Thwaites. I believe this is featured in the ABUMIP paper by Sainan Sun.

Reply: Thanks. We already said "The Weertman sliding parameterization is formulated assuming that temperate ice slides perfectly on a hard bed, ..." in this paragraph. We checked the ABUMIP paper, then add in the revision "It has been suggested that regularized Coulomb sliding parameterization is better suited for fast flowing areas such as Pine Island Glacier in the Amundsen Sea Embayment (Joughin et al., 2019)."

References:
Joughin, I., Smith, B.E., and Schoof, C.G.: Regularized coulomb friction laws for ice sheet sliding: application to pine island glacier, antarctica. Geophys. Res. Lett., 46(9), 4764–4771, 2019. doi: 10.1029/2019GL082526.

- I couldn't make sense of some of the equations in the method section due to what I expect is a PDF rendering error: some symbols appear as questionmarks. I am not sure where the error lies, but this made it impossible for me to assess these. It happened in Ln 142 – Ln 155

Reply: Sorry. The equations in Ln142 – Ln155 is the Glen's flow law. We will double check the pdf file, and make sure it will not happen next time.

- Ln 81: under which warming scenarios?

Reply: Yu et al. (2018) used different ice shelf melt scenarios: the ice shelf melt rate is parameterized as a function of ice shelf basal elevation and is set to zero above 150 m depth. The ice shelf melt rate linearly increases to a maximum at certain depth. They change the value of maximum to 40, 80, 120, 160 m/a. We change to "Yu et al. (2018) showed that the use of the Budd sliding parameterization produces more grounding line retreat and more VAF loss than with the linear Weertman sliding parameterization with then sub-shelf melt rate linearly dependent of ice shelf basal elevation and the maximum basal melt rate is larger than 80 m/a in simulations of Thwaites Glacier." in the revision.

- Ln 91: 'Has more complete physics' is a very broad statement and one that does not bring enough credit to other modelers groups in the world. I would rephrase this to something like 'Elmer/Ice has the option to run with the full stokes deviatoric stress tensor, which is an unique feature of this model'.

Reply: We do not intend to belittle other models, but the fact is that simplifications neglect some physics. We change it to "Since full-Stokes models consider all the components of the 3D deviatoric stress tensor, their physics is more complete than the simplified models (e.g., vertically integrated 'L1L2' approximation, 2D Shelfy-Stream stress balance approximation) that have been used in previous TG simulations..."

- Ln 91: what are 'the simplified models'?
Reply: We change it to "the simplified models (e.g., vertically integrated 'L1L2' approximation, 2D Shelfy-Stream stress balance approximation)"

- Fig 1: consider adding ice thickness as well, either as contour lines in the first plots or as a separate panel. Also, to remove the amount of numbers showed in these plots you can remove some of the axis labels. For example, in the top row, figure a,b and c share the same y-axis. Just showing it for figure a would be enough I think.

Reply: We improved Fig. 1 as below. We add ice thickness as a separate panel in Fig. 1a. The figures in each rows share the same y-axis. Another referee asked us to add an enlarged plot for each variable in Fig. 1.

- Fig 1c: I would suggest to use a logarithmic scale for this plot.

Reply: We used a logarithmic scale for plot (c).

- Fig 1e: Which reference did you use for these drainage basins?

Reply: We said in Fig. 1 caption that:

The solid black curve is the outline of Totten subregion, from MEaSUREs Antarctic Boundaries for IPY 2007-2009 from Satellite Radar, Version 2 dataset (Mouginot et al., 2017).

- Fig 1: the purple GL is almost invisible. Also, I would suggest to show the GL for the whole area, not just for Totten.

Reply: We have changed the grounding line to black. Also, we add enlarged subplots. The grounding line is clearer.

[Figure]

Figure 1. Bed elevation (a), surface elevation (b), surface ice flow speed (c), modelled basal temperature (g) and ice thickness (i) of Totten subregion and its surroundings. The solid black curve is the outline of Totten subregion, from MEaSUREs Antarctic Boundaries for IPY 2007-2009 from Satellite Radar, Version 2 dataset (Mouginot et al., 2017). The inland black curve is the grounding line The bed and surface elevations are from MEaSUREs BedMachine Antarctica, version 2 (Morlighem et al., 2020). The surface ice flow speed is from MEaSUREs InSAR-based Antarctic ice velocity Map, version 2 (Rignot and Scheuchl, 2017). Subglacial lake Vostok and Law Dome are marked in plot (a). The inset plot in plot (b) shows drainage basin divisions and the location of our domain (red curve) in Antarctica. The prescribed ice temperatures are taken from the output of the ice sheet model SICOPOLIS (Greve et al., 2020). (d)-(f) show the enlarged coastal region of (a)-(c), and (h) shows the enlarged coastal region of (g).

Fig 2: It is unclear to me which subfigure belongs where. The main blue discretization should be labeled (a), right? What is the use of showing the upper left red outline? Why are the observed velocities (are those ice surface velocities as well?) so much smaller than in Figure 1c? Also, you're scale bar looks off to me. Subpanel c should be about 400 km in width and maybe 500/600 km in lenght according to its own scalebar, while the red square in subpanel b is about 200 km in dimensions according to the bigger scale bar. I would also suggest to use a different color for the discretized elements in subfigure b. blue is already used in the two colormaps showing ice thickness and ice surface velocities.

Reply: Sorry to confuse you. We improved Fig. 2 as below. We reordered the subplots.

We changed the colorbar in plot (c) to be the same as Fig. 1c. But the color seems a bit darker because the background velocity map is covered with meshes. We corrected the scale bar. We use black for the discretized mesh.

[Figure]

Figure 2. The refined 2D horizontal domain footprint mesh (a). Box outlined in panel (a) is shown in detail and overlain with surface ice velocity in panel (b). The solid black curves in (b) represent the positions of the grounding line.

- Ln 137: which inverse method? And what target variable did you use?
Reply: We improved this sentence in the revision as:
We solve the Stokes equation and use a variational inverse method (Morlighem et al., 2010) as implemented in Elmer/Ice (Gagliardini et al., 2013; Gillet-Chaulet et al., 2012) to determine the basal friction coefficient in the linear Weertman sliding parameterization along with a stress enhancement factor.

- Ln 159: You could add a discussion on the not represented GIA. For 35 years it likely does not matter, but it would still be nice to read that and why.
Reply: We change in the revision: The bed for grounded ice is assumed to be rigid, impenetrable, and fixed over time since the ice sheet geometry change over decades is too small to affect lithosphere deformation.

- Ln 170: You could make Eq 6 and Eq 7 one equation, and just use m=1 for the linear cases. This saves space and does not require you to write C_lw.
Reply: We prefer to use two equations for the linear and nonlinear Weertman sliding, because we need use the different coefficient, $C_{LW}$ and $C_{NW}$ in the following conversion between the friction coefficients.

- Ln 172: In Eq 8 the vector u_b is used twice. I think the first one should be the magnitude u_b.
Reply: Yes, the first one should be the magnitude of u_b. We changed it.

- Ln 181: You could add Leguy et al 2014 and Leguy et al 2021, where they used a parameterization based on the height above floatation to mimic some hydrological connection for ice resting on bedrock below sea level.
Reply: Thanks. We add the two references.
Leguy, G. R., Asay-Davis, X. S., and Lipscomb, W. H.: Parameterization of basal friction near grounding lines in a one-dimensional ice sheet model, The Cryosphere, 8, 1239–1259, https://doi.org/10.5194/tc-8-1239-2014, 2014
Leguy, G. R., Lipscomb, W. H., and Asay-Davis, X. S.: Marine ice sheet experiments with the Community Ice Sheet Model, The Cryosphere, 15, 3229–3253, https://doi.org/10.5194/tc-15-3229-2021, 2021.

- Ln 190: In what quantifiable way does this alter your simulations?
Reply: We did the reference run (with Coulomb sliding law and sub-shelf melt rate with maximal value of 80 m/a) using two choices of effective pressure: 1) assuming effective pressure is hydrostatic; 2) setting the effective pressure as 5% of the overburden. We ran both for 35 years. We compare the two effective pressures at the end of the simulation, see plots below (Fig. S1). We found that the assuming perfect hydrostatic balance results in at least twice the effective pressure as assuming 5% of overburden near the grounding line, and an order of magnitude larger over the far inland region.

[Figure]

Fig. S1. Effective pressure set to 5% of overburden (a, b) and effective pressure assuming perfect hydrostatic connection and subglacial elevations (c, d) after 35 years simulation. Difference (e, f, i, j) and relative difference (g, h, k, l) between 5% of overburden and perfect connection at the beginning (e-h) and the end (i-l) of the 35 years simulation. The yellow and cyan curves represent the grounding line positions under 5% of overburden and perfect connection respectively after 35 years simulation. Note the different colorbar scales for (a, b) and (c, d).

We also compared the modelled surface velocity and grounding line position after the 35-year simulations using the two choices for effective pressures, see plots below (Fig. S2). The surface speed differs by $\pm$ 400 m/a mainly on the ice shelf and near the grounding line. There are significant differences in grounding line position after 35 years. The grounding line retreats more using the effective pressure under perfect hydrostatic connection. This grounding line position difference (Fig. S5) is similar to that between use of Coulomb sliding law and linear Weertman sliding law after 35 years (Fig. 6).

[Figure]

Fig. S5. Surface ice speed using 5% overburden effective pressures (a) and assuming perfect hydrostatic connection (b) after 35 years simulation. Difference (b − a) between (a) and (b) is shown in (c) and (d). The yellow and cyan curves represent the grounding line positions under 5% overburden and perfect hydrostatic connection respectively.

We change this sentence "This assumption alters simulation results significantly from those under the assumption of perfect hydrostatic connection" to
"This assumption decreases the effective pressure by an order of magnitude in most inland regions and halves it near the grounding line, compared with that under the assumption of perfect hydrostatic connection (Fig. S1). "
We add more text in the discussion.
"We compared the modelled surface velocity and grounding line position after 35 years of simulation using two choices of effective pressures: 1) assuming 5% of the overburden and 2) assuming perfect hydrostatic connection (Fig. S5). The surface speed differs by ±400 m/a mainly on the ice shelf and near the grounding line (Fig. S5). There are significant differences in grounding line position after 35 years. The grounding line retreats faster assuming perfect hydrostatic connection. This grounding line position difference (Fig. S5) is similar to that between use of Coulomb sliding law and linear Weertman sliding law after 35 years (Fig. 6)."

We also add Fig. S1 and Fig. S5 in the supporting information.

- Ln 191: What is a linear Weertman sliding inversion? Are there different inversions per basal friction law?

Reply: "The linear Weertman sliding inversion" means the inversion with the linear Weertman sliding parameterization. We only do inversion with the linear Weertman sliding law. We cannot do inversion for other types of basal sliding law directly. That is why we need convert the friction coefficients from linear Weertman sliding law to other sliding laws.

We changed it in the revision:

We apply the basal shear stress and basal sliding velocity field obtained from the inversion with the linear Weertman sliding parameterization to the other two sliding parameterizations, and estimate the sliding parameters there. In other words, we convert the local friction coefficient in the linear Weertman sliding parameterization to that in other sliding parameterizations.

- Ln 196-212: Am I understanding correctly here that, in the case of rewriting Weertman to Coulomb sliding, you could not find an exact solution for the free parameter in the coulomb sliding law and you had to revert to limits to find C_s with the inclusion of a smoothing term in Eq 14? If that is true, then, I like the elegance of this method but I am then not convinced that plugging the C_s you found in Eq 15 in Eq 8 will give you the same tau_b as Eq 7. Is that correct? And if so, can you then show that the deviation of tau_b is small and that it has no to little impact on a continuation simulation?

Reply: No, there is some misunderstanding. There are two free parameters in regularized Coulomb sliding, $A_s$ and $C_s$. We can find an exact solution for them. Once given $A_s$, we can calculate $C_s$. So we firstly find a reasonable expression of $A_s$, which is Eq. 14. Then we find the exact solution of $C_s$ (Eq. 15) which is dependent of $A_s$. We assume that the tau_b in Eq. 6 and Eq. 8 are equal. So $C_s$ is also dependent on $u_b$, $C_{LW}$, and $N$.

Plugging the $C_s$ in Eq. 15 into Eq. 8 will give us the same tau_b as Eq. 6 (note it is not Eq. 7). We do conversion from sliding coefficient in Eq. 6 to that in Eq. 8. The tau_b in Eq. 8 with $C_s$ in Eq. 15 is the same as the tau_b in Eq. 6. So there is no deviation of tau_b.

By the way, sorry, there is a typo in Eq. 15, the exponent "1-m" should be "m-1". We corrected it in the revision.

- Ln 220: what is shallow and what is deep water?

Reply: We add numbers here: shallow water (<500 m), deep water (>1000 m).

- Ln 225: Eq 16: where did you get the (non linear!) regression, and how well does it do in representing the values from the WAOM? A scatter plot between the WAOM basal melt rates versus draft depths, with this linear regression fitted through would make this clearer.

Reply: Sorry, the description was wrong. It is not a linear regression of ice shelf bottom depth and ice shelf basal melting rate. We removed the words "linear

regression". We did linear regression of $1/(d+d_0)$ and ice shelf basal melt rate, where $d$ is ice shelf bottom depth, and $d_0$ is a parameter to choose. We tried many values of $d_0$, and found that the linear regression of $1/(d+d_0)$ and sub-shelf basal melt rate has the best fit with the modelled result from WAOM when $d_0$=100.

We add in the revision:
We did linear regression between sub-shelf melt rate and $1/(d+d_0)$, where d is ice shelf bottom depth (unit: m), and $d_0$ is a tuned parameter. We found the best fit as follows with $d_0$ =100 m.

We add a scatter plot between the ice shelf bottom depth and WAOM modelled basal melt rate in the supplement (Fig. S2). We also plot the parameterization to account for the high melt rate in the deep water (Eq. 18), with $M_{max}$ = 40 m yr$^{-1}$, and the total sub-shelf basal melt (Eq. (21)) with water column thickness along the flowline FL1 as an example (Fig. S2).

We add in the revision:
The parameterizations for $M_d$, $M_e$ with $M_{max}$ = 40 m yr$^{-1}$ and the total sub-shelf melt with water column thickness along a flowline are illustrated in Fig. S2.

[Figure]

Fig. S2. Basal melt parameterization. Positive value of basal melt rate are melting and negative freezing rates. Whole Antarctic Ocean Model (WAOM v1.0; Richter et al., 2022) simulated basal melt rate (points) - which are far lower than spatially averaged satellite observational rates - see Section 3.3. The blue curve shows Eq. (17) tuned to WAOM modelled sub-shelf basal melt rates ($d_0$ = 100 m). The red curve is parameterized to account for the observed high melt rates in the deep water (Eq. (18)

with a $M_{max}$ = 40 m yr$^{-1}$). The sub-shelf melt is set to zero for ice shelf bottom depth less than 80 m (Eq. (20)). The black curve is the total sub-shelf basal melt (Eq. (21)) with water column thickness along the flowline FL1.

- Ln 225: Eq 16: I am also not sure about the addition of d_0 here. In my view, d is always nonzero if there is ice present in a grid element. If there is no ice present, one does not need to calculate the melt rates. Adding 100 is quite substantial, say your draft depth is also 100 m (it typically is between 0 and 1000 m, order of magnitude), adding 100 to your denominator for the sake of preventing zeros will alter your results significantly. Can you not remove this d_0?

Reply: We need d_0 for two reasons: (1) ensure the denominator term is nonzero.; (2) we did linear regression between ice shelf basal melt rate and $1/(d+d_0)$, we tried many values of $d_0$ and found that $d_0$=100 m gives the best fit. In the revision, we change it to "We did linear regression between sub-shelf melt rate and $1/(d+d_0)$, where $d$ is ice shelf bottom depth (unit: m), and $d_0$ is a tuned parameter. We found the best fit as follows with $d_0$=100 m".

- Ln 245 – 255: I am missing a discussion of the physical interpretation of the S_i and S_w values, just stating that the reflect the influence of cavity geometry and avoid numerical instability is in my opinion not enough. How does the cavity geometry influence the melt rates, and how is that reflected by S_w? Same for the numerical instability.

Reply: Regarding impact of S_i on numerical stability: A vanishingly thin ice shelf causes the tetrahedral elements to have a very high horizontal to vertical length scale, which increases the likelihood of instability. S_i prevents melting the ice shelf once the draft passes Z_s (i.e. no melting for thin ice).

Regarding impact of S_w on stability: A high step change in forcing across the grounding line (whether basal drag or ocean induced melt) causes problems for stability, but this is a main focus of Gladstone 2017. You could look there for more details. S_w prevents high melting right next to the grounding line, so reduces that step change.

Regarding cavity geometry influencing melt rates: a very thin water column restricts circulation (hence S_w). Observations of ocean induced melt always have peak rates at depth and very low melt rates for thinner parts of the ice shelf, hence S_i.

We add in the revision

And $S_w$ approaches 0 when the water column thickness goes to zero near the grounding line, capturing the influence of cavity geometry on melt rate as a very thin water column restricts circulation. A high step change in forcing across the grounding line (basal drag or ocean induced melt) causes problems for stability (Gladstone et al., 2017). $S_w$ prevents high melting right next to the grounding line, so reduces that step change.

A vanishingly thin ice shelf causes the tetrahedral elements to have a very high horizontal to vertical length scale, which increases the likelihood of instability. $S_i$ is an ice-shelf depth-scaling parameter, used to prevent melting the ice shelf once the draft passes $|z_{i0}|$ given by ...

- Fig 3: consider adding the observations to this figure as well.
Reply: We assume the referee means satellite estimates of basal melt rate by "observation". since we already have an enlarged plot for modelled sub-shelf melt rate with the maximum value of 160 m yr$^{-1}$ in plot (e), we replace the small figure of modelled sub-shelf melt rate with the maximum value of 160 m yr$^{-1}$ with the satellite estimates by Adusumilli et al. (2020) in plot (d).

Reference:
Adusumilli, S., Fricker, H. A., Medley, B. et al. Interannual variations in meltwater input to the Southern Ocean from Antarctic ice shelves. Nat. Geosci. 13, 616–620 (2020). https://doi.org/10.1038/s41561-020-0616-z

[Figure]

**Figure 3.** Spatial distribution of modelled sub-shelf melt rate with the maximum value of **(a)** 20 m yr$^{-1}$, **(b)** 40 m yr$^{-1}$, **(c)** 80 m yr$^{-1}$, **(e)** 160 m yr$^{-1}$ with coloured contours indicating water column thickness under the ice shelf with 100 m intervals. Plot **(d)** shows the satellite-based estimates (Adusumilli et al., 2020).

- What is inverted for is clear to me, the basal friction parameters C. However, how this is done (nudging, data assimilation) and with what as target (ice thickness, velocity) is not clear to me. Are you using the inversion results of Gladstone et al 2019? The reference here runs to an empty DOI, so I could not check what inversion you are using. Immediately after that you state the resolution of that simulation to be 4-40 km. Your simulation uses up to 900 meter resolution, how are you interpolating the inverted values without introducing model drift? What's more (and possibly more important) you are using the newest Bedmachine

dataset of Morlighem, 2020, Gladstone et al 2019 could not have used this, so they inverted a friction parameter using a different bedrockheight dataset. Also, your temperature profile from SICOPOLIS is different. You have to convince me now that you can take inverted fields from a different model run with different input datasets, approximations and parameterizations, and without problems use it in your own setup. You mention another study at the end of this paragraph (Gladstone and Wang, 2022) where some explanation is given but for Pine Island instead of Totten. I would suggest to add the equations you use from this paper and copy them to your study.

Reply: Sorry that you find an empty DOI for Gladstone et al. (2019), but it is the correct doi and it is there. Maybe it is caused by some internet problem. Anyway, we attach the paper as supporting information with the revision.

The initial value of basal friction coefficient, $\beta$, is from Gladstone et al. (2019), in which they used Bedmap 2 geometry. Our mesh is finer than that used in Gladstone et al. (2019). We bi-linearly interpolated the basal friction coefficient from Gladstone et al. (2019) to our mesh as an initial estimate of $\beta$. Then we use the inverse method to adjust the spatial distribution of basal friction coefficient $\beta$ to minimize the mismatch between the magnitudes of the simulated and observed surface velocities. The method is the variational inverse method (Morlighem et al., 2010) implemented in Elmer/Ice (Gagliardini et al., 2013; Gillet-Chaulet et al., 2012), and the target is surface velocity. Then we got our own optimal value of $\beta$ for the BedMachine geometry we used.

The inverse method procedure is the same as in Gladstone and Wang (2022). We add more equations in the revision.

Our temperature profile is taken from the output of the ice sheet model SICOPOLIS (Greve et al., 2020). Then we interpolated their value to our geometry.

- Ln 266: A diagnostic simulation tells me that you already did some kind of spinup or initialization. I would mention here your spinup procedure, and mention the inversion procedure as well.
Reply: The diagnostic simulation here means a steady state simulation, which can be viewed as part of the initialization. We change the word "diagnostic" to "steady state". But it does not include any inversion yet. The initialization includes an initial steady state simulation, surface relaxation, inversion for basal friction, and inversion for enhancement factor.

We re-organized this paragraph as
The following simulations are carried out in series as part of the initialization procedure. We firstly do a steady state simulation with $E_\eta=1$, and the linear Weertman sliding parameterization, expressing the sliding coefficient by $C_{\mathrm{LW}}=10^\beta$.

where $\beta$ is from the output of a whole Antarctic ice sheet inversion for the year 2015 (Gladstone et al., 2019), and linearly interpolated from their mesh to the finer mesh in this study. Then we relax the free surface of the domain by a short transient run of 1 year with a small time-step size of 0.1 year to reduce the non-physical spikes in the initial surface geometry (Zhao et al., 2018; Gladstone and Wang, 2022; Wang et al., 2020). Taking the results from surface relaxation, we use the variational inverse method (Morlighem et al., 2010) to adjust the spatial distribution of basal friction coefficient $\beta$ to minimize the mismatch between the magnitudes of the simulated and observed (Fig. 1c; Table 1) surface velocities.

- Ln 272: 'the inverse method'. What inverse method? Please specify.
Reply: It is the variational inverse method (Morlighem et al., 2010). We add it as above.

- Ln 275: If I get it correctly, you target surface ice velocities in your inversion, first with basal friction and then with the viscosity. It would be nice to read something here on this serial approach, why not in parallel? And what was the effect of this extra inversion step with the flow enhancement factor? Can the basal friction inversion alone not give the right ice surface velocities? Also, what was done by Gladstone et al 2019?
Reply: Yes, we target surface ice velocities in our inversion, first with basal friction and then with the viscosity, i.e. the enhancement factor. It is in series. We cannot invert for two variables as the same time. The inversion for viscosity is the inversion for enhancement factor. It is not an extra inversion step.

Some studies, e.g., Gladstone et al. (2019), use basal friction inversion alone to give the right ice surface velocities. We also consider enhancement factor because the viscous response of polar ice can be strongly anisotropic. On the coastal area, due to the large contrast of the stress regimes for the grounded part and for the ice shelf, the ice anisotropy induces an apparent hardening of the ice up to a factor 10 when ice moves from grounded to floating (Ma et al., 2010). The enhancement factor is often used to account for anisotropy effects. We add in the model description "The enhancement factor is often used to account for anisotropy effects (Ma et al., 2010). "

In Gladstone et al. (2019), they used BEDMAP2 geometry. The main target is minimizing the mismatch between modelled and observed velocities. Also a short surface relaxation removes extreme non-physical geometry artefacts and brings the shelves into floatation. The outputs are a spatially varying basal drag coefficient, 3D temperature field, and relaxed geometry. They carried out the following simulations in series as part of the initialization procedure (inversions tune basal friction parameter to MEASURES2 velocities):

1. Surface relaxation. 20 timesteps with dt = 0:001 a.
2. Inversion with relaxed geometry and constant temperature T = 20 C.
3. Steady state temperature simulation using the flow field from 2.
4. Inversion with the new temperature field from 3.
5. Thermo-mechanically coupled steady state temperature-velocity calculation using basal sliding coefficient from 4.
6. Further inversion using the latest temperature field from 5.
7. 10-year surface relaxation with a variable (increasing) timestep.

References:

Ma, Y., Gagliardini, O., Ritz, C., Gillet-Chaulet, F., Durand, G., and Montagnat, M.: Enhancement factors for grounded ice and ice shelves inferred from an anisotropic ice-flow model, Journal of Glaciology, 56, 805–812, https://doi.org/10.3189/002214310794457209, 2010.

- Ln 286: Why is regularized coulomb the most physically sound? Provide references.
Reply: We add the references: Tsai et al., 2015; Joughin et al., 2019.

- Ln 290: SMB has already been mentioned, consider removing it here.
Reply: Done.

- Ln 291: I would argue that, if computational expenses are too high when running Full Stokes, shift to a faster approximation (Hydrostatic, Blatter-Pattyn or for example DIVA) and run further into the future. 35 years is short to make statements about sensitivities to basal friction, maybe the simulations will start to deviate as soon as the grounding line retreats further (e.g. after 100-200 years). Or converge to some steady state upstream. Doing 35 year simulations in my opinion is particularly usefull when making state-of-the art projections of glacier retreat, with forcing from CMIP models. For sensitivity studies like this one, I would recommend to run longer.
Reply: Computational expenses are high for using Full-Stokes model. We compared the short simulation between full-Stokes and Blatter-Pattyn model, however, we found large difference in modelled ice velocity. Switching from Stokes to Blatter-Pattyn halves the speed in the shelf. Therefore, it is not useful to run Blatter-Pattyn model for longer times.

Since 35 years is short to make statements about sensitivities to basal friction, we tried our best to run it longer to 2100. We finished the runs with three different basal friction parameterizations and one ice shelf basal melt rate (max=80 m/a). We updated the results and Fig. 5-8 .

- Ln 296: you do not keep beta fixed right? You rewrite them to fit with other friction parameterizations. Please state so. Also, please make clear which beta (the one from Gladstone et al 2019 or one obtained after your own relaxation) was used.
Reply: We firstly obtain the optimal beta in the inversion with linear Weertman sliding

parameterization. Then we convert it to the coefficients in other sliding parameterizations. Then we keep them fixed in the prognostic simulations. We already said in section 3.4 "With the modelled basal shear stress and basal sliding velocity, we convert the basal drag coefficient from the linear Weertman parameterization to those representing non-linear Weertman and regularised Coulomb parameterizations using the method described in Section 3.2."

We made it more clear in section 4.1 "We obtain the inverted optimal spatial distribution of basal sliding coefficient exponent (Fig. 4a) in the linear Weertman parameterization and the stress enhancement factor (Figs. 4b-c) for the initial year 2015, and keep them fixed in all the prognostic simulations".

- Fig 4: the grounding line is very hard to see, consider changing the colors and/or the thickness. Also I would like to see the observed grounding line position next to the modelled one to asses how well your model performs.

Reply: We change the color of grounding line in Fig. 4. We also add enlarged plots for each variable in Fig. 4. The initial grounding line is determined by the ice sheet geometry from Bedmachine data using the flotation condition. The Bedmachine data is from merged observational data from several years. Hence we already show the observational grounding line. The inversion is a steady state simulation which does not change the initial (observed) grounding line.

[Figure]

**Figure 4.** Spatial distribution of modelled **(a)** basal sliding coefficient $\beta$, **(b)** stress enhancement factor $E_\eta$, **(c)** basal shear stress $\tau_b$, **(d)** basal effective viscosity, and their corresponding enlarged ones (e-h). The solid yellow (a-c; e-g) or cyan (d, h) curves represent the grounding line and the solid white curves in (a-d) show modelled basal speed contours of 10 m yr$^{-1}$.

- Ln 311: Why does the ice shelf decelerate? I would expect some speedup due to the

loss of buttressing due to the loss of ice shelf thickness.

Reply: We change it to "In the reference run, ice shelf thickness upstream of the grounding line decreases significantly, and the ice velocity speeds up upstream of the grounding line but decelerates near the ice front of the ice shelf over the 85-year simulation".

- Fig 5: a difference plot would be more informative here, since the visual difference between beginning and end of the simulation is hard to see.

Reply: We update Fig. 5 to add the difference plot and extend the simulation result to 2100.

- Fig 5 c-f: I appreciate the honesty when saying that the vertical lines are Python artefacts, but I would still like them to be removed before publication.

Reply: We found the problem and fixed it. There are no artifacts in our new plots.

[Figure]

Figure 5. Surface velocity at the beginning of the initial year 2015 (a) and the end of the year 2100 (b) in the reference run. The surface velocity difference (2100 minus 2015) is shown in (c). The grounding line positions in the year 2015, 2040, 2060, 2080 and 2100 are shown in (a-c). Pink and purple solid lines in (a) and (b) represent flowlines FL1 and FL2 as labelled. The solid color portions of the figures show the ice flow velocity (upper colorbar) profiles along FL1 (c, d) and FL2 (e, f) in the reference run in the initial year 2015 (d, f) and the end year 2050 (e, g), with bedrock in dark grey and seawater in blue. The geometry change of TG is marked with colored solid lines for the years 2015, 2040, 2060, 2080 and 2100. The vertical elevations are exaggerated by a factor of 25.

- Ln 327-333: this conclusion, that there are various grounding line retreats for different sliding laws, is not what I got from reading your abstract in which you mentioned that the basal sliding law did not matter.
Reply: We update Fig. 6a with the result of longer years to 2100. We checked the abstract, to make the conclusion consistent.

- Ln 339: this is not neccesarely the case: less sliding and particular coulomb sliding will make it easier for ice to flow from far upstream to the grounding line, preventing the thinning at the grounding line.
Reply: We change it to "The grounding line retreats more using the regularised Coulomb sliding parameterization than with the nonlinear Weertman sliding parameterization, which might because the effective pressure used in the regularised Coulomb is reduced as the ice thins, leading to more basal sliding and faster ice speed.

Moreover, the value of $\chi$ (Eq. 9) is below 1 in the fast flowing region from our posterior estimate, showing a reduced basal friction and enhanced basal sliding (Eq. 16) compared with a true regularised Coulomb sliding parameterization."

- Ln 353 – 355: I do not agree here. First, the magnitudes of the spatial velocity differences might imply something on horizontal shear and its derivatives, but why is that relevant? Also, if you want to show the spatial variability in the shear stresses, why do you not plot the shear stresses themselves? But the most important point: there are multiple other approximations that take either horizontal or vertical derivatives of the shear stresses into account, or combinations of them. You can pick Hybrid SIA+SSA for example, or the Depth Integrated Viscosity Approximation (DIVA), or Blatter-Pattyn, or the Hydrostatic Approximation. Those will all resolve the quantities you want to detect, with less computational expenses. This does not justify the need for a Full-Stokes model, and if you can only run for 35 years with Full Stokes, I would strongly suggest to run longer with a less computational heavy approximation, or at the very least rephrase and rethink why you chose Full Stokes in the first place.
Reply: As you said, the ice speed difference (surface minus basal) implies something about horizontal shear, or the basal drag. In fact, we found that the ice speed difference variation is very similar to the basal drag variation. In addition, the pattern of difference between the surface and basal ice speed implies high spatial gradients in basal velocity,

We agree we can show the basal drag directly. So we add a separate panel to show the basal drag.

As we mentioned earlier, "Switching from Stokes to Blatter-Pattyn for a single iteration in the prognostic run halves the speed in the shelf, which suggests the necessity of using the full-Stokes model". We extended the prognostic runs in the

Sliding parameterization group to the year 2100.

- Ln 355: Despite what differences? Also, you just argued that there is a huge difference in grounding line response (10 km vs 1 km), now you are writing the opposite. From figure 6, I conclude that there is quite a difference in grounding line retreat when using a different sliding law, contrasting your abstract.
Reply: Sorry for the confusion. We remove this sentence.

- Ln 355: Why is this clearly controlled by the topography? I cannot see this in Figure 6.
Reply: We mean they all mainly retreat along the eastern side, but little on the western side and southern side. That is a minor point. We remove this sentence.

- Ln 370: this conclusion seems to be a bit obvious, that melt rates directly influence the cavity thickness. Whats more interesting is the relation between sliding law and cavity thickness. Can you quantify this effect? Why does another friction parameterization lead to different ice shelf cavity thickness?
Reply: The sub-shelf cavity change is dominated by the sub-shelf melt rate. Considering the the relation between sliding law and cavity thickness, we can see from Fig. 7 and Fig. 8 that in the case of nonlinear Weertman sliding law, the sliding speed is slower than in the other two cases. This causes less grounding line retreat, hence thicker ice shelf near the grounding line. It also could cause less advection of ice through the shelf, hence thinner ice further downstream in the shelf.

We also note that the basal topography along FL 1 is prograde sloping bed with large slope upstream of grounding line, but the basal topography along FL 2 has more variations in slope and the slopes are smaller than that along FL1. Hence the ice is more stable along FL 1 than along FL2. The influence of different basal sliding parameterizations on sub-shelf cavity thickness is more obvious along FL2 than along FL1.

We change in the revision to:
The change of sub-shelf cavity thickness is dominated by sub-shelf melt rates, although different basal sliding parameterizations could yield different retreat of grounding line position, hence different sub-shelf cavity thickness near the grounding line. The influence of different basal sliding parameterizations on sub-shelf cavity thickness is negligible in prograde sloping bed with large slope upstream of the grounding line such as FL1 (Fig. 7), and obvious at locations with relatively small and variable slope such as FL2 (Fig. 8).

- Fig 6 and fig 7: I see much more grounding line retreat difference in Fig 6 compared to Fig 7, why is that?
Reply: Sorry, maybe it is because the plotting scale was wrong in Fig. 2c. Fig. 2c and Fig. 6 have the same scales. We corrected the plotting scale in Fig. 2c and added the

plotting scale in Fig. 6 in the updated plots.

Reply: There is no seasonal cycle in our simulation. The plot was made using the results of every timestep. So there is some oscillation. We change to use annual result to make the plot. Then it is smooth.

Reply: We only take it as an initial guess of basal friction coefficient. Then we use the inverse method to adjust the spatial distribution of basal friction coefficient to minimize the mismatch between the magnitudes of the simulated and observed surface velocities. Therefore, the basal friction coefficient from Gladstone et. al (2019) has no influence on our modelled result.

Reply: It is low. But we get the ratio from subglacial hydrology modelling result (Dow et al., 2020). And we compared the simulations with two choices of effective presure in the reply to Ln 190.
.
We read Leguy et al. (2014). They proposed a simple function for the effective pressure, $N(p)$, depending on a parameter $p$, that accounts for connectivity between the subglacial drainage system and the ocean: $N(p) = 0$ at the grounding line when $p > 0$, and $N(p)$ is the ice overburden when far from the grounding line. It is a nice function. But it faces a key challenge of choosing realistic values of $p$ for the study domain. We would like to try that in a future study. We added some text in the discussion.

"Leguy et al. (2014) proposed a function for the effective pressure, N(p), depending on a parameter p, that accounts for connectivity between the subglacial drainage system and the ocean. It could produce a smooth transition between finite basal friction in the ice sheet and zero basal friction in the ice shelf, but it faces a key challenge of choosing realistic values of p for the study domain."

Reference:
Leguy, G. R., Asay-Davis, X. S., and Lipscomb, W. H.: Parameterization of basal friction near grounding lines in a one-dimensional ice sheet model, The Cryosphere, 8, 1239–1259, https://doi.org/10.5194/tc-8-1239-2014, 2014

- Overall: the short timescales are not discussed in this section, and its possible effect

on the simulations. Barnes and Gudmundsson 2022 and Brondex et al 2017 and Brondex et al 2019 all conducted longer simulations, so you might find similar or non-simular results if you extend your simulations to match their lengths, typically 100-300 years.
Reply: We extend our simulations to the year 2100 with three different basal sliding laws. We find similar results as that by the year 2050. We updated the figures and text in the revision.

- Ln 445: I would not ask you for general statements, but what I would like to read is your thoughts on the fact that shapewise (in a basal friction versus basal velocity plot) the non-linear Weertman and coulomb sliding law look more like each other, than the coulomb sliding law and the linear Weertman. Why are then the modelled ice sheet responses so similar between the linear Weertman and Coulomb sliding law?
Reply: That is a good point. The regularised Coulomb law can provide a continuous transition between Coulomb friction regime and Weertman type friction regime, and the basal drag instantaneously switches from Coulomb friction at low effective pressure to Weertman type friction at high effective pressure. It depends on the parameter $\chi$. We showed in our manuscript that in the limit of small $\chi$, we recover a non-linear Weertman law (Eq. (11)), and in the limit of large $\chi$, we obtain a Coulomb-type of friction (Eq. (13)). Also note that, Eq. (10) is equivalent to Coulomb-type of friction with a factor:

$$\tau_b = \left(\frac{\chi}{1+\chi}\right)^{1/m} C_S N.$$

The factor $\left(\dfrac{\chi}{1+\chi}\right)^{1/m}$ is about 0.6 if $\chi$=0.275, 0.8 if $\chi$=1, and 0.97 if $\chi$=10. It means a reduced basal friction than that in a true regularised Coulomb law.

The regularised Coulomb law has a complicated form, and there are two free parameters, A_s and C_s. the parameter $\chi$ depends on A_s, C_s, basal velocity and effective pressure. In our approach, we firstly decide the form of A_s, then we calculate C_s using the prescribed A_s. Then we find the resulting value of $\chi$ is below 1 over the domain, and from 0.3 to 1 in the fast-flowing region with spatial variability and change with time, which means the basal friction in the fast-flowing region is 60%-80% of that in a true Coulomb regime. We also tried to use different form of A_s, but it is hard to get $\chi$ as a good transition from very small value in the slow-flowing region to a large (>10) value in the fast-flowing region.

During the work we also tried another approach: we firstly give a constant value of $C\_s$, then we calculate $A\_s$ using the prescribed $C\_s$. Then we hope to have both large and small value of $\chi$, i.e. both Weertman type regime and Coulomb regime. However, the assumption of constant value of $C\_s$ is not reasonable, and the transition of resulting $\chi$ is still not good. How to design reasonable parameters ($A\_s$ and $C\_s$) requires further investigation as a longer-term project.

Therefore, we still use the first approach, but we say clearly in the text that it is a regularised Coulomb law with a reduced basal friction .

- Ln 447: 'the maximal basal melt rates are'.
Reply: Done.

- Ln 449: this seems contradictory: is it consistent or are there large differences?
Reply: It is not contradictory. We found that the retreat occurs mainly along the eastern and southern grounding zones. This is consistent with previous studies. But our modelled retreat distance is different from previous studies. We rewrote the sentence as: "The modelled grounding line retreats when maximal basal melt rate is $\geqslant$ 40 m yr$^{-1}$, and mainly along the eastern and southern grounding zones. This is consistent with previous studies, although with different retreat distance."

- Ln 452: 'Melt' has a typo
Reply: Corrected.

- Ln 455 - 456: Earlier on I read this: 'The sensitivity of sub-shelf cavity thickness to basal sliding parameterization varies spatially', which seems to contradict this statement. Which is true?
Reply: We correct the sentence here:
The sensitivity of sub-shelf cavity thickness to basal sliding parameterization depends on basal topographic slope and ice thickness.

- Ln 465: consider publishing your scripts to make the figures and datasets of your simulations as well.
Reply: Okay.

---

## Author Comment (AC2)

Referee's comments are in blue, our reply in black, quotes in the revised manuscript in red.

**Manuscript Review**
"Sensitivity of Totten Glacier dynamics to sliding parameterizations and ice shelf basal melt rates"
Yiliang Ma, Liyun Zhao, Rupert Gladstone, Thomas Zwinger, Michael Wolovick and John C. Moore

**Summary:**
Ma et al. present an analysis of the effects of three different basal sliding parameterisations and four different ice-shelf basal melt parameterisations on the evolution of Totten Glacier. To do this they use the full-Stokes model capabilities of ElmerttIce and perform simulations over a 35-year period from 2015-2050. They find that grounding line retreat occurs when the maximum value in the basal melt parameterisation is greater than 40 m/a. They also find that the linear Weertman sliding law generates the most grounding line retreat, closely followed by the regularised Coulomb sliding law, with the non-linear Weertman sliding law producing the least grounding line retreat and mass loss over the 35-year simulation period.

The paper is well-written, and the results provide additional insights into the effects of commonly used basal sliding parameterisations on the dynamics of an important glacier in East Antarctica when simulated with a full-Stokes model. However, I have a major point that I would like to see addressed before publication, relating to the presentation and discussion of the full-Stokes model results.

**Major Comment:**
My concern is with the presentation and analysis of the modelled ice velocities in Figures 5, 7 and 8, and in the text in lines 349-354.

Panels (c) and (d) of Figure 7 (and 8 to some extent) show large oscillations in the difference between the surface and basal ice speed along a flowline upstream of the grounding line. In some experiments, this difference approaches 800 m/a. This difference must be due to vertical shearing in the ice, and this result implies that over three-quarters of the total flow comes from vertical shearing in these fast-flowing ice stream regions. The fact that the difference between surface and basal ice speed then drops close to zero a few kilometres upstream/downstream also suggests that this large vertical shearing quickly disappears as a factor in the dynamics, and the flow is then dominated by basal slip without any clear variation in the surface or basal topography.

These estimates for vertical shearing in the ice are very large (typical values would be < 100 m/a) and don't appear to be physically plausible. It is also unexpected that there could be such profound changes in the dominant mechanism for flow over a few

kilometres along a flowline without any appearance of this in the overall surface speed (as shown by the much smoother curves in panels (a) and (b) of Figure 7 when compared to panels (c) and (d)).

Reply: Firstly, we think the difference between the surface and basal ice speed comes from horizontal shear rather than vertical shear.

Secondly, we do not agree that "These estimates for vertical shearing in the ice are very large (typical values would be < 100 m/a) and don't appear to be physically plausible."

Panels (a) and (b) of Figure 7 are surface velocity, they are smooth because the surface topography is relatively smooth. But there could be large variation in basal velocity if the basal topography is rough. The basal velocity depends on ice temperature and also basal topography. For instance, a pinning point could slow down the ice velocity. Therefore, the difference between the surface and basal ice could have large variation. The largest difference between surface and basal ice speed, 800 m/a, happens with ice thickness of ~2 km and surface speed of 1 000 m/a. We think it is plausible with steep basal slope and thick ice. The oscillations correspond to changes of basal topography, basal friction coefficient and basal drag. We can also see the stripes pattern in the modelled basal friction coefficient upstream the grounding line in Fig. 4. The variation in basal friction coefficient is related with the variation in basal velocity (Fig. S6).

Thirdly, we do not agree that "The difference between surface and basal ice speed then drops close to zero a few kilometres upstream/downstream ..."
The difference between surface and basal ice speed does not drop to zero in the 50 km upstream along either FL1 or FL2, see Fig. 7 and Fig. 8. The difference between surface and basal ice speed is zero downstream the grounding line is because it is the floating ice shelf there, and the surface and basal ice speed in an ice shelf is the same.

Could there have been an issue with the post-processing of the model data? The artefacts in Figure 5 – that the authors attribute to plotting in Python – also lead me to this as a possibility. Whilst it is hard to tell definitively, it appears that the oscillations in basal and surface speed that are clear in Figures 7 and 8 are also visible in the ice velocities plotted in panels (c) and (d) of Figure 5 and are attributed to plotting artefacts there. The pronounced gradients in the surface and basal ice speeds (in both the vertical and horizontal dimensions) shown by the stripes of different shades of green appear to be in the same locations as some of the largest oscillations in Figure 7 panels (c) and (d).

Reply: Now we figured out the plotting issue in Python and fixed it. There is no artifact in the updated figures Fig. 5. The oscillations in Figure 7 and 8 is not relevant

to Fig 5. The oscillations in Figure 7 and 8 are physically plausible, as we answered in the earlier reply.

Could the authors please verify that these results are not due to an error in the post-processing of the model data? Perhaps visualising the data in ParaView and comparing it with their Python generated plots might reveal potential discrepancies. A spatial map of the basal ice velocity would also be useful in understanding what is going on in the model output.

Reply: Yes, we verified it. The oscillations in Figure 7 and 8 are relevant to Fig 5. The oscillations in basal velocity correspond to those in basal friction coefficient and basal drag. You can see the stripes pattern in both basal shear stress and basal friction coefficient (Fig. 4). Anyhow, we made a spatial map of the basal ice velocity, see the plot below.

[Figure]

Fig. S6. Modelled basal velocity in the steady state (the initial year 2015).

If this is indeed the behaviour of the full-Stokes model in this region, then I think that the authors need to expand much more on these results. This would also require a physical explanation for the readers to understand what mechanisms could be driving such large rates of vertical shearing in the ice and the large variations in its contribution to the flow over just a few kilometres in the horizontal dimension, without any expression in the overall surface ice speed.

Reply: The ice is stress-balanced by the gravitational driving force, the basal friction and extensional stress divergence term. The gravitational driving force depends on ice thickness and basal slope. It could change greatly where the ice thickness or basal elevation changes dramatically. We infer the basal drag coefficient such that the modelled surface velocity matches the observed. The surface velocity changes smoothly. Hence, the basal drag should change greatly to offset changing gravitational driving force. Therefore, we can see stripes pattern in the inferred basal drag or basal friction coefficient.

We add more words for physical explanation for Fig. 7 and 8 in the revision:

The pattern of difference between the surface and basal ice speed implies high spatial gradients in basal velocity, and reflects the basal drag which must change in response to steep basal slopes or with large variations in ice thickness to balance the gravitational driving force.

Reference:
Morlighem, M., Rignot, E., Seroussi, H., Larour, E., Ben Dhia, H., and Aubry, D.: Spatial patterns of basal drag inferred using control methods from a full-Stokes and simpler models for Pine Island Glacier, West Antarctica, Geophys. Res. Lett., 37, L14502, https://doi.org/10.1029/2010GL043853, 2010.

**Minor Comments:**
Line 92: Can you explain why you expect to gain more information on basal processes from the full Stokes model compared to the range of approximations that are available and open used? It would be good to have more justification for the benefits of your use of full-Stokes given the fact that it limits your experiments to just 35 years.
Reply: Morlighem et al. (2010) compared the inferred basal drag from full Stokes (FS), shelfy stream (SSA) and the Blatter-Pattyn (BP) models, and found that near the glacier grounding line, SSA and BP exhibit a high basal drag (80 kPa), while the basal drag inferred from FS is less than 10 kPa. Therefore, FS is essential to infer a correct pattern of basal drag and to capture all higher order stresses in the grounding line region. They suggest that near the grounding-line of ice streams, treating ice flow with the complete physics of FS is essential.

We change the sentence here to:
"Since full-Stokes models consider all the components of the 3D deviatoric stress tensor, their physics is more complete than the simplified models (e.g., vertically integrated 'L1L2' approximation, 2D Shelfy-Stream stress balance approximation) that have been used in previous TG simulations. Furthermore, full-Stokes models have been suggested as essential to infer a correct pattern of basal drag and to capture all higher order stresses near the grounding-line of ice streams (Morlighem et al., 2010). Therefore, we expect to gain more insight into basal processes using a full-Stokes model.

We also extend the simulations with full-Stokes model to the year 2100 with three different sliding parameterizations in the revision.

Figure 1: Could you show these plots zoomed-in on the area of interest (as in Figure 2 (c) and elsewhere)? As you have a 35-year experiment and only see limited grounding line retreat, much of the model domain is not of interest to your results, and by zooming out the reader loses much of the detail in bed elevation or flow speed that is important.

Reply: We made zoomed-in plots in Fig. 1.

[Figure]

Figure 1. Bed elevation (a), surface elevation (b), surface ice flow speed (c), modelled basal temperature (g) and ice thickness (i) of Totten subregion and its surroundings. The solid black curve is the outline of Totten subregion, from MEaSUREs Antarctic Boundaries for IPY 2007-2009 from Satellite Radar, Version 2 dataset (Mouginot et al., 2017). The inland black curve is the grounding line The bed and surface elevations are from MEaSUREs BedMachine Antarctica, version 2 (Morlighem et al., 2020). The surface ice flow speed is from MEaSUREs InSAR-based Antarctic ice velocity Map, version 2 (Rignot and Scheuchl, 2017). Subglacial lake Vostok and Law Dome are marked in plot (a). The inset plot in plot (b) shows drainage basin divisions and the location of our domain (red curve) in Antarctica. The prescribed ice temperatures are taken from the output of the ice sheet model SICOPOLIS (Greve et al., 2020). (d)-(f) show the enlarged coastal region of (a)-(c), and (h) shows the enlarged coastal region of (g).

Figure 2: It's not clear what is gained by showing the inset in panel (b) here, I would consider removing it.

Reply: We removed panel (b).

Line 190: I would be interested to know what impact this choice has on your results, either discussed here or in the discussion section. It seems important given your use of a pressure-dependent sliding law and the impacts you hint at here.

Reply: We did the reference run (with Coulomb sliding law and sub-shelf melt rate with maximal value of 80 m/a) using two choices of effective pressure: 1) assuming effective pressure is hydrostatic; 2) setting the effective pressure as 5% of the overburden. We ran both for 35 years. We compare the two effective pressures at the end of the simulation, see plots below (Fig. S1). We found that the effective pressure as 5% of the ice overburden is an order of magnitude smaller than that assuming perfect hydrostatic balance over the far inland region, and at least halves that assuming perfect hydrostatic balance near the grounding line.

[Figure]

Fig. S1. Effective pressure set to 5% of overburden (a, b) and effective pressure assuming perfect hydrostatic connection and subglacial elevations (c, d) after 35 years simulation. Difference (e, f, i, j) and relative difference (g, h, k, l) between 5% of overburden and perfect connection at the beginning (e-h) and the end (i-l) of the 35 years simulation. The yellow and cyan curves represent the grounding line positions under 5% of overburden and perfect connection respectively after 35 years simulation. Note the different colorbar scales for (a, b) and (c, d).

We also compared the modelled surface velocity and grounding line position after 35 year simulations using the two effective pressures, see plots below (Fig. S2). The surface speed differs by $\pm 400$ m/a mainly on the ice shelf and near the grounding line. There are significant differences in grounding line position after 35 years. The grounding line retreats more using the effective pressure under perfect hydrostatic connection. This grounding line position difference (Fig. S5) is similar to that

between use of Coulomb sliding law and linear Weertman sliding law after 35 years (Fig. 6).

[Figure]

Fig. S5. Surface ice speed using 5% overburden effective pressures (a) and assuming perfect hydrostatic connection (b) after 35 years simulation. Difference (b − a) between (a) and (b) is shown in (c) and (d). The yellow and cyan curves represent the grounding line positions under 5% overburden and perfect hydrostatic connection respectively.

We change this sentence "This assumption alters simulation results significantly from those under the assumption of perfect hydrostatic connection" to "This assumption decreases the effective pressure by an order of magnitude in most inland regions and halves it near the grounding line, compared with that under the assumption of perfect hydrostatic connection (Fig. S1). "

We add more text in the discussion.
"We compared the modelled surface velocity and grounding line position after 35 years of simulation using two choices of effective pressures: 1) assuming 5% of the overburden and 2) assuming perfect hydrostatic connection (Fig. S5). The surface speed differs by ±400 m/a mainly on the ice shelf and near the grounding line (Fig. S5). There are significant differences in grounding line position after 35 years. The grounding line retreats faster assuming perfect hydrostatic connection. This grounding line position difference (Fig. S5) is similar to that between use of Coulomb sliding law and linear Weertman sliding law after 35 years (Fig. 6)."

We also add Fig. S1 and Fig. S5 in the supporting information.

 Why do you need to have d0 = 100 in Equation 16? d should always be > 0 on an ice shelf. Even if, for numerical reasons, it can become 0, why use d0 = 100 to correct for that? Did you use (d – 100) as the value for the ice-shelf bottom depth in your linear regression to account for this constant?

Reply: We did linear regression of $1/(d+d_0)$ and ice shelf basal melt rate, where $d$ is ice shelf bottom depth, and $d_0$ is a parameter to choose. We tried many values of $d_0$, and found that the linear regression of $1/(d+d_0)$ and sub-shelf basal melt rate has the best fit with the modelled result from WAOM when $d_0=100$.

Line 279: You state that this initial state is representative of 2015, but the data sets used are mosaics whose data collection period spans decades (e.g. your Table 1 shows that the ice velocity is a mosaic of data from 1996-2016). Itim not sure that it is possible to state that your initial state is 2015 without using datasets timestamped to that year – especially for a region which has seen significant changes as you outline in your introduction.

Reply: We checked the user guide of BedMachine Antarctic dataset. It is said: "The data were collected between 01 January 1970 and 01 October 2019. The nominal year of this data set is 2015—the year of the reference surface digital elevation model."

Line 291: The short timescale of 35 years makes it more important to state the benefits of using full Stokes for these experiments to balance this limitation (see earlier point).

Reply: We extend the simulations with full-Stokes model to the year 2100 with three different sliding parameterizations in the revision. We also addressed the benefits of using full Stokes in the earlier point.

Figure 4: Again, I would prefer to see plots zoomed-in on the region of interest, as in panels (c) and (f) so that the details of the basal sliding and basal shear stress can be seen. The colour scale for the stress enhancement factor colour bar (white around 0) seems different to the one in the maps (grey around 0). Finally, I am not sure of the benefit of plotting the 10 m/a basal speed contours and suggest removing them.

Reply: We show zoomed-in plots for all the variables on the region of interest in Fig. 4. We checked the colorbar of the stress enhancement factor. The color around 1 (we guess you mean 1 rather than 0) is grey. We need the 10 m/a basal speed contours. We define the fast flow region where ice speed >10 m yr$^{-1.}$

[Figure]

**Figure 4.** Spatial distribution of modelled **(a)** basal sliding coefficient $\beta$, **(b)** stress enhancement factor $E_\eta$, **(c)** basal shear stress $\tau_b$, **(d)** basal effective viscosity, and their corresponding enlarged ones (e-h). The solid yellow (a-c; e-g) or cyan (d, h) curves represent the grounding line and the solid white curves in (a-d) show modelled basal speed contours of 10 m yr$^{-1}$.

Figure 5: See my main comments here for possible data issues, but if these are genuine artefacts of plotting in Python then they need to be corrected in updated plots. Reply: We figured out the plotting issue in Python and fixed it. There are no artifacts anymore.

Line 455: In the Results section (Line 366) you stated that the sub-shelf cavity thickness did depend on the basal sliding relation along FL2. This was surprising to me, and I would be interested to know what the physical mechanism linking the upstream conditions at the bed to the thickness of the iceshelf cavity could be, and also the strength of this relationship compared to the much more direct impact of different basal melt rates under the ice shelf. Please clarify this discrepancy here. Reply: We compare the impact on sub-shelf cavity (or the ice shelf bottom elevation) of basal sliding law with the much more direct impact of sub-shelf melt rates using the old Fig. 7 and Fig. 8 during the same period 2015-2050. The ice shelf bottom elevation is very similar along FL1 and there is slightly difference along FL2 in the year 2050 and more difference along FL2 in the year 2100 (see the updated Fig. 8 in the revision) because the grounding line positions are different. The sub-shelf cavity change is dominated by the sub-shelf melt rate.

We also note that the basal topography along FL 1 is prograde sloping bed with large slope upstream of grounding line, but the basal topography along FL 2 has more variations in slope and the slopes are smaller than that along FL1. Hence the ice is more stable along FL 1 than along FL2. The influence of different basal sliding

parameterizations on sub-shelf cavity thickness is more obvious along FL2 than along FL1.

We can see in the updated Fig. 8 that in the case of nonlinear Weertman sliding law, the sliding speed is slower than in the other two cases. This causes less grounding line retreat, hence thicker ice shelf near the grounding line. It also causes less advection of ice through the shelf, hence thinner ice further downstream in the shelf.

We change in the revision to:
The change of sub-shelf cavity thickness is dominated by sub-shelf melt rates, although different basal sliding parameterizations could yield different retreat of grounding line position, hence different sub-shelf cavity thickness near the grounding line. The influence of different basal sliding parameterizations on sub-shelf cavity thickness is negligible in prograde sloping bed with large slope upstream of the grounding line such as FL1 (Fig. 7), and obvious at locations with relatively small and variable slope such as FL2 (Fig. 8).

---

## Author Response (AR1)

Editor's comments are in blue, our reply in black, quotes in the revised manuscript in red.

**Dear Authors,**

Thank you for your replies. Both reviewers agree that this could be a valuable contribution to the field if you address their major concerns sufficiently. Some of your responses to the main points are rather brief and do not always completely resolve the underlying concerns. I am listing some of the issues below and would like to ask you for a more complete reply to the reviewer's concerns.

It is good that you picked up the reviewer's suggestion to run longer simulations, but you did not mention the results of these simulations in your response. Please add a discussion of these results and their implications in your revised manuscript.

Reply: We extended the simulations using the full-Stokes model to the year 2100 with three different sliding parameterizations but the same sub-shelf basal melt rate. Sorry, we did not mention the results in our previous rebuttal, although we updated related figures (Fig. 6, 7, 8, 9) in the previous revision.

We add more about these results and their implications in section 4.1 "4.1 Initial state and reference run result" of the revision as below:

We select two flowlines for analysis: FL1 along the main trunk, and FL2 on the eastern branch (Fig. 5) of TG. Modelled annual averaged grounded ice thinning rate from 2015 to 2100 measured from the 2100 grounding line position and extending 20 km further upstream is 2.14~2.35 m yr-1 along FL1 and 2.88~5.06 m yr-1 along FL2.

We add more about these results and their implications in section 4.2 "Sensitivity experiments results" of the revision as below:

Between 2015 and 2100, when applying the regularised Coulomb and linear Weertman sliding parameterizations, the surface ice flow speed exhibits an acceleration upstream of the grounding line, and the ice thickness upstream of the grounding line experiences a gradual decrease. This ice flow speed acceleration and ice thinning are more pronounced in the eastern grounding zones compared to the southern ones (Fig. S4, Fig. S5). Conversely, when the nonlinear Weertman sliding parameterization is employed, the surface ice flow velocity undergoes a slight deceleration in the vicinity of the grounding line, while it experiences a slight acceleration at locations further upstream (Fig. S4). The surface lowering is much smaller using nonlinear Weertman sliding parameterization than that using the regularised Coulomb and linear Weertman sliding parameterizations. More specifically, the surface ice speed change and ice thickness change along FL1 and FL2 from experiments in the sliding parameterization group are shown in Table 3, Figure 7 and 8.

Table 3. Surface ice speed change,  $\Delta v$ , and ice thickness thinning rate,  $\Delta h$ , from the year 2015 to 2100 (2100 minus 2015) at the 2100 grounding line position and 20 km further upstream, along FL1 and FL2 from experiments in the sliding parameterization group.

|                                |                     | U 1       | U 1 |           |          |
|--------------------------------|---------------------|-----------|------------|-----------|----------|
|                                |                     | Along FL1 |            | Along FL2 |          |
|                                | Sliding             | Grounding | 20 km      | Grounding | 20 km    |
|                                | Parameterizations   | Line 2100 | Upstream   | Line 2100 | Upstream |
| $\Delta v \text{ (m yr}^{-1})$ | Regularized Coulomb | 195       | 324        | 270       | 405      |
|                                | Linear Weertman     | 165       | 240        | 200       | 537      |
|                                | Nonlinear Weertman  | -31       | 18         | -39       | -34      |
| $\Delta h \text{ (m yr}^{-1})$ | Regularized Coulomb | 2.14      | 2.35       | 2.88      | 5.06     |
|                                | Linear Weertman     | 2.29      | 2.65       | 2.83      | 6.42     |
|                                | Nonlinear Weertman  | 0.59      | 0.91       | 0.70      | 0.79     |

From 2015 to 2100, when applying both the regularized Coulomb and linear Weertman sliding parameterizations, the grounding line along FL1 retreats at an approximate rate of 0.14 km yr-1 (Fig. 7). Along FL2, the retreat rates under these two parameterizations are approximately 0.27 km yr-1 and 0.42 km yr-1 respectively (Fig. 8). When the non-linear Weertman sliding parameterization is used, the grounding line retreats at a rate of around 0.04 km yr-1 from 2015 to 2050. However, after 2050, the retreat is negligible.

The simulation grounded area loss rate between 2015 to 2100 from the Sliding parameterization group experiments are heterogeneous in both time and space (Fig. 5, 6, 9a).

The modelled grounded area loss using nonlinear Weertman after the year 2050 is much less than that before 2050. The modelled grounded area loss rate using regularised Coulomb becomes slower from 2050 to 2075 while the glacier retreats over rumpled terrain, and re-accelerates after the year 2075. The modelled grounded area using linear Weertman decreases at a nearly constant rate becoming gradually slower in the last two decades.

However, the modelled VAF using the three basal sliding parameterizations decreases almost linearly over time (Fig. 9c), due mainly to the dynamic ice flow. The modelled VAF loss of the TG sub-basin from the Sliding parameterization group experiments is equivalent to global sea level rise of 5.67 mm, 5.48 mm and 3.29 mm over the period 2015-2050 and 16.35 mm, 15.97 mm and 7.34 mm over the period 2015-2100 using linear Weertman, regularised Coulomb and non-linear Weertman sliding parameterizations, respectively (Fig. 9c).

We also add two figures in Supporting Information as below.

Fig. S4. The surface ice speed change from the year 2015 to 2100 (2100 minus 2015). The solid black line represents the initial grounding line position in the year 2015. The orange, blue and pink curves represent grounding line positions in the year 2100 from experiments using regularised Coulomb (a), linear Weertman (b) and non-linear Weertman (c) in the Sliding parameterization group (Table 2).

Fig. S5. The ice thickness change from the year 2015 to 2100 (2100 minus 2015). The solid black line represents the initial grounding line position in the year 2015. The orange, blue and pink curves represent grounding line positions in the year 2100 from experiments using regularised Coulomb (a), linear Weertman (b) and non-linear Weertman (c) in the Sliding parameterization group (Table 2).

The statement that BP leads to a 50% reduction compared to FS is a bit thin and I would encourage you to elaborate more on the model differences in case of BP and FS. It is also not clear whether you run BP until the year 2100 as well.

Reply: Sorry. We found a mistake in the setup of BP simulation that showed a 50% reduction. We corrected it. We only ran steady state iteration using the BP model, restarting from the steady state result of FS model. We did not run the BP model until the year 2100.

We do more comparisons between BP and FS as shown below. We compare the surface ice speed and basal ice speed between BP and FS, see the figure below.

The figure shows: Both models reveal similar surface and basal velocity patterns. The ice speed simulated using BP is several hundred meters per year faster than that

simulated using FS near the grounding line, but several hundred meters per year slower at the ice shelf, especially near the ice front.

Fig. S7. The simulated surface (the first row) and basal (the second row) ice speed in steady state simulation using full-Stokes model (a, d) and Blatter-Pattyn model (b, e) restarting from the steady state result of full-Stokes model, and their difference (c, f) which is Blatter-Pattyn minus full-Stokes.

We found a reference, Rückamp et al. (2022), which compares the ice dynamics using full-Stokes (FS) and Blatter-Pattyn (BP) approximation. They found that both models simulate similar surface velocity patterns, but the BP solution exhibits higher flow speeds in the fast flow region and slower in the slow flow region. The differences between FS and BP increase with higher velocities, and are stronger when using a power-law friction than a linear friction law. The velocity difference between FS and BP we find in our results is consistent with Rückamp et al. (2022).

**We add in the revision:**

We ran a steady state simulation using the Blatter-Pattyn model, restarting from the steady state result of the reference run using the full-Stokes model and using the same basal sliding parameterization as in the reference run. We found that both models reveal similar surface velocity patterns (Fig. S7). The ice speed simulated using Blatter-Pattyn is several hundred meters per year faster than that simulated using full-Stokes near the grounding line, but several hundred meters per year slower at the ice shelf, especially near the ice front. This is consistent with the finding in Rückamp et al. (2022), which compares the full-Stokes and Blatter-Pattyn models applied to the Northeast Greenland Ice Stream. They also found that the discrepancies between full-Stokes and Blatter-Pattyn increase with higher velocities, and are stronger when using a power-law friction than a linear friction law. The speed difference in the vicinity of the grounding line is significant because marine ice sheet behaviour is largely controlled by feedbacks (involving both ocean-induced melt and basal resistance)

close to the grounding line. The Blatter-Pattyn simulation predicting faster flow speed in this region would cause differences in the evolution over time of the system, hence we chose to focus on full-Stokes simulations for the current study. A comprehensive investigation into the time evolving impacts of the various approximations to the Stokes equations on evolution of the Totten Glacier would be valuable, but is not the focus of the current study where we use a complete representation of stresses.

**Reference:**

Rückamp, M., Kleiner, T., and Humbert, A.: Comparison of ice dynamics using full-Stokes and Blatter–Pattyn approximation: application to the Northeast Greenland Ice Stream, The Cryosphere, 16, 1675–1696, https://doi.org/10.5194/tc-16-1675-2022, 2022.

The major concern of reviewer 2 should be addressed more thoroughly. In your revision you merely state that you think that the differences in velocities derive from horizontal shear and that the ripples/oscillations in the differences between surface and basal speed are due to either rough terrain or basal friction coefficient. In figure 7 there is no sign of strong bedrock roughness/variability, and you don't provide an explanation for the ripples in the basal friction coefficient which are derived from inversion. A more thorough explanation of these model results would be welcome. In your response you state: "The variation in basal friction coefficient is related with the variation in basal velocity". While it is of course true that variability in the basal friction coefficient drives variability in basal sliding, no explanation is provided what the causes of the variations in basal friction are.

Reply: We agree this question by referee 2 is good point. Sorry we did not reply to it thoroughly.

We add more discussion in this revision as below. Although there is no direct evidence of strong bedrock roughness/variability, we provide an explanation for the ripples in the basal friction coefficient which are derived from inversion. The truth is, that inversion results do not directly give us the cause of any given pattern in the basal friction, they only allow us to infer that the pattern is there. Ripples, oscillations, or "ribs" have been found in other inverse models of both Greenland and Antarctica, starting with Sergienko and Hindmarsh (2013). We add a brief discussion of the ribs below:

Rib-like patterns in basal friction have been found in inversions in both Antarctica and Greenland (Sergienko and Hindmarsh, 2013; Sergienko et al., 2014; Wolovick et al., 2023). These ribs in the basal friction mimic, but do not exactly follow, similar rib-like patterns in the gravitational driving stress (Wolovick et al., 2023). Sergienko and Hindmarsh (2013) posited that basal traction ribs form from coupled instabilities in the till-water-ice system, and supported that supposition with process modeling which produced broadly similar rib-like patterns as those seen in their inverse model. However, strictly speaking, an inversion cannot tell us the definitive cause of any

given structure in the inverted field, it can only inform us that the structure exists. Wolovick et al. (2023) found that rib-like structure in the basal drag was sensitive to the choice of regularization in the inversion, which should be expected for short-wavelength features. However, they also found that the ribs were present in both their base case inversion with the best optimal corner lambda value, as well as in their best combined basal drag map. We did not perform a full L-curve analysis in this paper, but we did test values of  $10^3$ ,  $10^4$ ,  $10^5$ ,  $10^6$ ,  $10^8$ ,  $10^{10}$ , for the regularization parameter  $\lambda_{\beta}$  in the inversion, choosing  $10^3$ , which is relatively small, and hence amenable to more variable basal friction fields. If we had chosen a larger lambda value, it is likely we would have seen fewer ribs in our inverted result. However, the fact that similar ribbed structure has been seen in many different inverse models applied to many different geographic regions- and that ribbed structure is also seen in the gravitational driving stress, which does not depend on any inverse model results- suggests that at least some of this structure must be genuine.

**Reference:**

Sergienko, O. V., & Hindmarsh, R. C. A.: Regular patterns in frictional resistance of ice-stream beds seen by surface data inversion. *Science*, *342*(6162), 1086–1089, 2023. Sergienko, O. V., Creyts, T. T., & Hindmarsh, R. C. A.: Similarity of organized patterns in driving and basal stresses of Antarctic and Greenland ice sheets over extensive areas of basal sliding. *Geophysical Research Letters*, *41*(11), 3925–3932, 2014.

Wolovick, M., Humbert, A., Kleiner, T., and Rückamp, M.: Regularization and L-curves in ice sheet inverse models: a case study in the Filchner–Ronne catchment, *The Cryosphere*, 17, 5027-5060, https://doi.org/10.5194/tc-17-5027-2023, 2023.

I agree with the reviewer that close to the grounding line and upstream (a few kilometers) velocity differences appear to be zero (at least your figure 7 suggests this for the "initial" case). It would probably help if you highlighted the grounding line position in panels a-f as well to make this clearer.

Reply: There was some misunderstanding of the location where the velocity differences appear to be zero. What we thought is the place where the velocity is stably close to zero, rather than the isolated zeros in the large oscillations. Now we see the referee means the isolated zeros in the large oscillations. Then we agree with the reviewer that there are large oscillations of velocity difference within tens of kilometers upstream the grounding line, and the velocity differences are close to zero at places a few kilometers upstream the grounding line.

**We add in the revision**

The oscillations in the difference between the surface and basal ice speeds near the grounding line and upstream are caused by the rib-like pattern of basal friction and basal velocity. The difference between surface and basal speeds drops to zero, which means the basal friction approaches zero in the fast flowing trunks upstream of the grounding line (Figs. 7 and 8). The fast-flowing ice streams are supported by shear

margins or isolated sticky regions through long-distance stress transmission (Wolovick et al., 2023).

We highlight the grounding line positions in all panels of Figure 7 and Figure 8.

Figure 7. Surface ice speed (a, b), ice speed difference (c, d; surface minus basal), basal drag (e, f) ice thickness change (g, h), and ice-sheet profiles (i, j) along FL1 in the initial year 2015 (black solid line) and the year 2100 (coloured solid line) from experiments (Table 2) in the Sliding parameterization group (left column) and the year 2050 (coloured solid line) in the Melt rate group (right column). grounding line positions are marked with black vertical dashed line for the year 2015 and coloured vertical dashed line for the year 2100 in the Sliding parameterization group (left column) and the year 2050 (coloured solid line) in the Melt rate group (right column). The figure is shaded to show the geometry of the TG along the flow line in 2015, where dark grey is the bedrock, light blue the ice shelf and dark blue seawater. The elevations are exaggerated by a factor of 25.

Figure 8. The same as Figure 7 but for FL2.

It is good that you fixed the python issues, but I am surprised by your statement that the oscillations in figure 7/8 are not relevant in figure 5. Figure 5 shows a transect and the ice speed along this transect through depth. Thus, both the surface as well as the basal ice speed is shown in Figure 5 and illustrated in Figure 7/8 in a different way. In your next reply to the reviewer you state that the variations in Figure 5 are indeed the ones you see in Figure 7/8. I would suggest that you address the reviewers' comments more clearly both in your response as well as in the revised manuscript especially regarding the question why the inversion produces the undulating basal friction coefficient patterns in absence of strong bedrock relief variability and no evident drastic changes in ice thickness/surface slopes.

Reply: What we mean in the earlier rebuttal is that the oscillations in Figure 7/8 are not relevant with the vertical lines in the old Figure 5, which are Python artifacts. We see your meaning now. There are oscillations in basal friction coefficient and speed difference in Fig. 7/8, hence there should be also oscillations in the ice speed along the transect of flowlines in Fig. 5. In fact, there is spatial variability in ice speed along the flowlines in Fig. 5, but it is not very noticeable due to the log scale of the colorbar.

We do not know why the inversion produces the undulating basal friction coefficient patterns in absence of strong bedrock relief variability and no evident drastic changes in ice thickness/surface slopes. The inversion results do not directly give us the cause

of any given pattern in the basal friction, they only allow us to infer that the pattern is there. Ripples, oscillations, or "ribs" have been found in other inverse models of both Greenland and Antarctica, starting with Sergienko and Hindmarsh (2013). We add in the revision:

Rib-like patterns in basal friction have been found in inversions in both Antarctica and Greenland (Sergienko and Hindmarsh, 2013; Sergienko et al., 2014; Wolovick et al., 2023). These ribs in the basal friction mimic, but do not exactly follow, similar riblike patterns in the gravitational driving stress (Wolovick et al., 2023). Sergienko and Hindmarsh (2013) posited that basal traction ribs form from coupled instabilities in the till-water-ice system, and supported that supposition with process modeling which produced broadly similar rib-like patterns as those seen in their inverse model. However, strictly speaking, an inversion cannot tell us the definitive cause of any given structure in the inverted field, it can only inform us that the structure exists. Wolovick et al. (2023) found that rib-like structure in the basal drag was sensitive to the choice of regularization in the inversion, which should be expected for shortwavelength features. However, they also found that the ribs were present in both their base case inversion with the best optimal corner lambda value, as well as in their best combined basal drag map. We did not perform a full L-curve analysis in this paper, but we did test values of 103, 104, 105, 106, 108, 1010, for the regularization parameter  $\lambda_{\beta}$  in the inversion, choosing 103, which is relatively small, and hence amenable to more variable basal friction fields. If we had chosen a larger lambda value, it is likely we would have seen fewer ribs in our inverted result. However, the fact that similar ribbed structure has been seen in many different inverse models applied to many different geographic regions- and that ribbed structure is also seen in the gravitational driving stress, which does not depend on any inverse model results- suggests that at least some of this structure must be genuine.

Please also make sure that you address the minor comments of both reviewers conclusively. Some examples below:

I agree with the reviewer's comment that the formulation "their physics is more complete than the simplified models" is not necessary here. It is enough to state that Full Stokes models include all stress components as opposed to approximations such as SIA/SSA etc.

Reply: Okay. We remove "their physics is more complete than the simplified models".

As suggested in the review process regarding the missing bedrock response to changes in ice load: it would be good to include some references which estimate the effect of e.g. 1D ELRA models or fully fledged GIA models on decadal to centennial ice sheet responses.

Reply: We add some references in the revision.

The bed for grounded ice is assumed to be rigid, impenetrable, and fixed over time since the ice sheet geometry change over decades is too small to affect lithosphere deformation (de Boer et al., 2014; Coulon et al., 2021).

**References:**

de Boer, B., Stocchi, P., and van de Wal, R. S. W.: A fully coupled 3-D ice-sheet-sealevel model: algorithm and applications, Geosci. Model Dev., 7, 2141-2156, https://doi.org/10.5194/gmd-7-2141-2014, 2014.

Coulon, V., Bulthuis, K., Whitehouse, P. L., Sun, S., Haubner, K., Zipf, L., & Pattyn, F.: Contrasting response of West and East Antarctic ice sheets to glacial isostatic adjustment, J. Geophys. Res.-Earth Surf., 126, e2020JF006003, https://doi.org/10.1029/2020JF006003, 2021.

Please quantify the costs for the Full Stokes, BP simulations either in the methods or supplements.

Reply: We add the costs in the revision as below.

These simulations were executed on a server equipped with 2 Intel(R) Xeon(R) Platinum 9242 CPUs operating at 2.30GHz and 384GB memory. Restarting from the steady state result using full-Stokes model in the reference run, the steady state simulation using Blatter-Pattyn model takes 1 minute and 11 seconds, while one timestep forward iteration using full-Stokes model with linear Weertman sliding parameterization takes 15 minutes.

As per the reviewers' comments please use the term "significant" only where it applies to a quantifiable statistical estimate.

Reply: We checked it. We replace "has significant effect on" to "has marked effect on". We replace "significant future grounded ice loss" to "notable future grounded ice loss". We replace "have a significant impact on" to "have an important impact on". We replace "show significant different transient behaviour" to "show notably different transient behaviour", etc.

Best regards,

Johannes Sutter

---

## Referee Report (RR1)

**Additional comments on 'Sensitivity of Totten Glacier dynamics to sliding parameterizations and ice shelf basal melt rates'.**

Dear Yiliang Ma et al,

Thank you for this much improved manuscript, and taking many of our concerns into account. I would like to recognize the hard work (and computational expenses) the authors put in extending their simulations to 2100.

I have some additional minor comments, with the most emphasis on convincing me that your setup does not show, or shows very little, model drift. If there is drift (not unexpected when doing a data-assimilation initialization) I would like to see it quantified and discussed in how it affects your results and conclusions.

Ln 34: I would add (Schoof, 2012) as a reference to MISI here

Ln 126: You use time-varying SMB fields from CESM2 – CMIP6 ensemble, right? If so, please specify that you use 1995- 2100 CESM2 anually varing SMB fields (and, is space allows, include a plot to show what the SMB looks like over the modelled period)

Ln 146: 'Version 9.0.' should be mentioned before the citation and after 'Elmer/Ice'

Ln 154 – 165: I am still seeing poorly-loaded question marks for the symbols used in your equations. I've added a screenshot below here. I double checked if this was due to my version of Adobe Acrobat and it was not. Please make very sure during the typesetting process that the symbols are represented correctly.

Ln 189: try to avoid starting sentences with a parameter symbol. You could change this to 'The parameter A\_s...'

Ln 189: 'is the sliding parameter without cavitation' does not make it clear to me what the parameter does, what it's value is, or what it's units are. A general comment on this

section: consider adding a table at the beginning or end of this chapter where you list all parameters, their units, and their values used.

Ln 205 – Ln 210: the rewriting of free parameters in friction laws has been done by (Brondex et al., 2017), please refer to that paper here in this paragraph as well.

Ln 210 – Ln 234: The same questionmarks appear as shown in the screenshot earlier. Please pay special attention when compiling the final manuscript

Ln 265 – Ln 279: I would write this part directly after Ln 244 – Ln 250. In that way, the flow of text would be more logical e.g. 'In shallow water we use...', 'in deep water we use ...' and then 'The WAOM estimates...'

Ln 298 – 315: could you quantify or show if there is any model drift in the final initialization you used? Is this setup robust? I would like to see a line added to Fig 9 representing a simulation starting from 2015 and remaining unforced, or, if you do not want to change your main text too much, see it added to the supplementary materials. I would like to see that your setup is not drifting from 2015 – 2100 and that you can attribute all changes in TG's geometry to changes in basal friction and/or basal melting.

Ln 298 – 315: related to that, it would be good state in this section what happens with the basal melt rate during the initialization procedure of Gladstone and Wang (2022) and during your 1 year relaxation (I assume they are kept constant). In that way you can state explicitely (related to the previous comment): 'during our initialization we used basal melt rates from ... /shown in supplementary material, and when starting a continuation run we immediately switch to ... . Keeping the melt rates and the basal friction constant and running forward to 2100 did not introduce model drift, as shown in Fig (S) ...'

Ln 336: Please avoid starting sentences with parameter symbols

Ln 337: 'show a rib like pattern'

Ln 337: 'regions'

Ln 338: 'it does not directly give us the cause'.

Ln 339: The dot after the references appears as red to me. Is this a track-change left in the document?

Ln 340: I would replace 'unity' with just 'one/1'

Ln 342: Xi varies from 0.3 - 0.5 (what units?) and then increases to 0.3 - 0.5? This repetition of values seems like an error to me

Ln 393: You could also argue that the rib-like pattern is caused by other errors in the ice sheet modelling process and/or by the use of non-coherent datasets, as described by (Berends et al., 2023)

Ln 407: 'might be because'

Ln 495: Did you also run with the same 1-year relaxation period?

Ln 510: Cluster specific details are not necessary here in my opinion. Stating that Blatter Pattyn is 15 times faster would be sufficient.

521: Or, the structures arise as compensating errors to other missing processes or uncertainties in datasets, see my comment on Ln 393

Ln 571: This is not true, the rewriting was done first by (Brondex et al., 2017)

**References**

- Berends, C. J., Van De Wal, R. S., Van Den Akker, T., & Lipscomb, W. H. (2023). Compensating errors in inversions for subglacial bed roughness: same steady state, different dynamic response. *The Cryosphere*, *17*(4), 1585-1600.
- Brondex, J., Gagliardini, O., Gillet-Chaulet, F., & Durand, G. (2017). Sensitivity of grounding line dynamics to the choice of the friction law. *Journal of Glaciology*, 63(241), 854-866.
- Schoof, C. (2012). Marine ice sheet stability. *Journal of Fluid Mechanics*, 698, 62-72.

---

## Author Response (AR2)

Referees' comments are in blue, our reply in black, quotes in the revised manuscript in red.

**Referee 1**

Additional comments on 'Sensitivity of Totten Glacier dynamics to sliding parameterizations and ice shelf basal melt rates'.

Dear Yiliang Ma et al,

Thank you for this much improved manuscript, and taking many of our concerns into account. I would like to recognize the hard work (and computational expenses) the authors put in extending their simulations to 2100.

I have some additional minor comments, with the most emphasis on convincing me that your setup does not show, or shows very little, model drift. If there is drift (not unexpected when doing a data-assimilation initialization) I would like to see it quantified and discussed in how it affects your results and conclusions.

Reply: We have carefully considered the issue of model drift. But we do not think it is meaningful to keep the setup unforced for model drift evaluation. We need to force the model with the fixed present-day climate forcings, i.e, SMB and sub-shelf melt rate. We checked the SMB in different years, and found it does not vary much from 2015 to 2100. The estimates for present-day sub-shelf melt rate varies in earlier studies. It seems that our sub-shelf melt rate parameterization with maximal value of 80 m/a is close to the present-day estimates.

Therefore, a drift control simulation is almost the same setup as the reference simulation, C\_m80, which is using Coulomb sliding law, SMB and sub-shelf melt rate with maximal value of 80 m/a. So, a model drift simulation is not needed. Furthermore, our paper is more a sensitivity study than future projection, so quantifying drift in the initialized model is not very important.

Ln 34: I would add (Schoof, 2012) as a reference to MISI here Reply: Done.

Ln 126: You use time-varying SMB fields from CESM2 – CMIP6 ensemble, right? If so, please specify that you use 1995-2100 CESM2 anually varing SMB fields (and, is space allows, include a plot to show what the SMB looks like over the modelled period) Reply: Yes, we used time-varying SMB fields modelled by CESM2 for the period 2015-2100. We specify 'time-varying' in the revision. There is no space to add SMB plot in the main text. But we add it in the supplementary, also shown as below.

Fig. S1. The averaged SMB field for the period 2015-2100 simulated by CESM2.

Ln 146: 'Version 9.0.' should be mentioned before the citation and after 'Elmer/Ice' Reply: Corrected.

Ln 154 - 165: I am still seeing poorly-loaded question marks for the symbols used in your equations. I've added a screenshot below here. I double checked if this was due to my version of Adobe Acrobat and it was not. Please make very sure during the typesetting process that the symbols are represented correctly.

where T is the deviatoric stress tensor, p is the isotropic pressure,  $\rho_i$  is ice density, g is the acceleration due to gravity (0, 0, -9.81) m s-2, and v is ice velocity. The effective viscosity,  $\Phi$  for the ice-rheology,

$$\tau = 2\eta \dot{x}$$
, (3)

given by Glen's flow law (Cuffey and Paterson, 2010),

$$\eta = \frac{1}{2} \left( E \cdot A(T) \right)^{-\frac{1}{n}} \varepsilon_n^{\frac{1-n}{n}} \tag{4}$$

7

is dependent on ice temperature via the flow rate factor A(T), defined by an Arrhenius equation. We use an exponent of n=3.  $\mathcal{E}$  denotes strain rate and the term  $\mathcal{E}_g = \sqrt{tr(\mathcal{E}^2)/2}$  is the effective strain rate. The ice temperature distribution comes from the output of ice sheet model SECOPOLIS (Greve et al., 2020). We introduce  $\Phi_{\mathbf{e}} = \Phi^{-1/(2)\Phi}$ , to obtain

$$\eta = \frac{1}{2} E_{\eta}^{2} \cdot A(T)^{-\frac{1}{u}} \frac{1-u}{\varepsilon_{g}^{u}}$$
(5)

which ensures  $\eta > 0$  for any value of  $\bullet$ . We refer  $\bullet$  to as the 2D vertically constant stress enhancement factor which is set initially to unity everywhere, indicating no stress enhancement, before performing the viscosity inversion. The enhancement factor is often used to account for anisotropy effects (Ma et al., 2010). Values of  $\bullet < 1$  indicate softer and faster deforming ice, whereas higher values,  $\bullet > 1$ , mean stiffer ice. Reply: Okay. It looks fine in our pdf. We retype the symbols and hope it can be displayed normally in your computer next time.

Ln 189: try to avoid starting sentences with a parameter symbol. You could change this to 'The parameter A s....'

Reply: Done.

Ln 189: 'is the sliding parameter without cavitation' does not make it clear to me what the parameter does, what it's value is, or what it's units are. A general comment on this section: consider adding a table at the beginning or end of this chapter where you list all parameters, their units, and their values used.

Reply: We change 'the sliding parameter without cavitation' to 'the sliding parameter in the absence of cavitation', see Gagliardini et al. (2007). Its unit is  $m \ a^{-1} \ Pa^{-3}$ . We use a complicated formula to calculate it, see Eq. (15). So it is hard to know its value before simulation. According to our simulation results, it has a very wide value range from  $10^{-16}$  to  $10^2 \ m \ a^{-1} \ Pa^{-3}$ .

We add a table to list all the parameters and their units.

Table 2. List of parameters used in Section 3.2 and their units.

| Parameter | Description                                        | Unit                   |
|-----------|----------------------------------------------------|------------------------|
| $C_{LW}$  | friction coefficients of linear Weertman           | m -1 yr Pa  |
| $C_{NW}$  | friction coefficients of nonlinear Weertman        | $m^{-1/3} yr^{1/3} Pa$ |
| $C_s$     | the maximum value reached by $\tau_b/N$            | No unit                |
| $A_s$     | the sliding parameter in the absence of cavitation | $m yr^{-1} Pa^{-3}$    |
| q         | friction law exponent, $q \ge 1$ , we take $q=1$   | No unit                |

Ln 205 – Ln 210: the rewriting of free parameters in friction laws has been done by (Brondex et al., 2017), please refer to that paper here in this paragraph as well.

Reply: We cite that paper here in the revision.

Ln 210 – Ln 234: The same question marks appear as shown in the screenshot earlier. Please pay special attention when compiling the final manuscript

Reply: Okay.

Ln 265 – Ln 279: I would write this part directly after Ln 244 – Ln 250. In that way, the flow of text would be more logical e.g. 'In shallow water we use...', 'in deep water we use...' and then 'The WAOM estimates...'

Reply: The WAOM estimates is what we used for the shallow water. We get the point of the referee's suggestion. We reordered the paragraphs to make the shallow water estimate and deep water estimate closer. We move the different estimates from previous studies to the end of this section, and add a sentence for transition: The modelled sub-shelf melt rates, with varying maximal values, cover most of the estimates from previous studies.

Ln 298 – 315: could you quantify or show if there is any model drift in the final initialization you used? Is this setup robust? I would like to see a line added to Fig 9 representing a simulation starting from 2015 and remaining unforced, or, if you do not want to change your main text too much, see it added to the supplementary materials. I would like to see that your setup is not drifting from 2015 – 2100 and that you can attribute all changes in TG's geometry to changes in basal friction and/or basal melting. Reply: We think the definition of model drift is the differences over time in the modelled glacier behavior that would occur if we employed present-day climate forcings. For instance, Seroussi et al. (2019) estimated the drift in the control run, in which climate forcing is assumed to be similar to present-day conditions. So, we do not think it is meaningful to run an unforced model drift evaluation. For a drift control experiment, we need to force the model with the fixed present-day climate forcings, i.e., SMB and sub-shelf melt rate. The sub-shelf melt rate applied to the ice sheet can however change due to, e.g., variations in ice extent during the simulation.

We checked the SMB in different years, and found it does not vary much from 2015 to 2100. Our sub-shelf melt rate parameterization with maximal value of 80 m/a is close to the estimates of present-day sub-shelf melt rate.

Therefore, a drift control simulation is almost the same setup as the reference simulation (and the reason we chose it as a reference), C\_m80, using Coulomb sliding law, SMB and BMB with maximal sub-shelf melt rate of 80 m/a. Furthermore, our paper is a sensitivity study rather than future projection, so quantifying drift in the initialized model is not as important as the differences between simulations.

**Reference:**

Seroussi, H., Nowicki, S., Simon, E., Abe-Ouchi, A., Albrecht, T., Brondex, J., Cornford, S., Dumas, C., Gillet-Chaulet, F., Goelzer, H., Golledge, N. R., Gregory, J. M., Greve, R., Hoffman, M. J., Humbert, A., Huybrechts, P., Kleiner, T., Larour, E., Leguy, G., Lipscomb, W. H., Lowry, D., Mengel, M., Morlighem, M., Pattyn, F., Payne, A. J., Pollard, D., Price, S. F., Quiquet, A., Reerink, T. J., Reese, R., Rodehacke, C. B., Schlegel, N.-J., Shepherd, A., Sun, S., Sutter, J., Van Breedam, J., van de Wal, R. S. W., Winkelmann, R., and Zhang, T.: initMIP-Antarctica: an ice sheet model initialization experiment of ISMIP6, The Cryosphere, 13, 1441–1471, https://doi.org/10.5194/tc-13-1441-2019, 2019.

Ln 298 - 315: related to that, it would be good state in this section what happens with the basal melt rate during the initialization procedure of Gladstone and Wang (2022)

and during your 1 year relaxation (I assume they are kept constant). In that way you can state explicitly (related to the previous comment): 'during our initialization we used basal melt rates from ... /shown in supplementary material, and when starting a continuation run we immediately switch to ... . Keeping the melt rates and the basal friction constant and running forward to 2100 did not introduce model drift, as shown in Fig (S) ...'

Reply: The initialization procedure includes 1) steady state simulation with a basal sliding coefficient from the output of a whole Antarctic ice sheet inversion for the year 2015 (Gladstone et al., 2019); 2) 1 year relaxation; 3) Inversion for basal drag; 4) Inversion for viscosity enhancement factor.

We do not need to prescribe basal melt rate during the steady state simulation or inversions, because they are diagnostic simulations for a fixed ice sheet geometry. We set the basal melt rate equals 0 during the 1 year relaxation because we do not want to change the ice shelf geometry and 1 year is too short to have a large effect.

The basal melt rate we used for surface relaxation is different from that used in Gladstone and Wang (2022). In the surface relaxation simulation of Gladstone and Wang (2022), the basal mass balance under the ice shelf uses the Ice Sheet Model Intercomparison Project (ISMIP6) "local quadratic melting parameterisation". To avoid confusion, we remove Gladstone and Wang (2022) from the reference in the text about initialization. In the revision, We change to "Then we relax the free surface of the domain by a short transient run of 1 year with a small time-step size of 0.1 year without any sub-shelf melt rate to reduce the non-physical spikes in the initial surface geometry".

Ln 336: Please avoid starting sentences with parameter symbols

Reply: We add words in front of symbols.

Ln 337: 'show a rib like pattern'

Reply: Done.

Ln 337: 'regions'

Reply: Done.

Ln 338: 'it does not directly give us the cause'.

Reply: Done.

Ln 339: The dot after the references appears as red to me. Is this a track-change left in the document?

Reply: Yes, we change it to black.

Ln 340: I would replace 'unity' with just 'one/1'

Reply: Okay. We change 'unity value' to '1'.

Ln 342: Xi varies from 0.3 - 0.5 (what units?) and then increases to 0.3 - 0.5? This repetition of values seems like an error to me

Reply: Xi has no unit. What we wrote is: Xi increases to 0.3-5 rather than 0.3-0.5. So it is not repetition of values.

Ln 393: You could also argue that the rib-like pattern is caused by other errors in the ice sheet modelling process and/or by the use of non-coherent datasets, as described by (Berends et al., 2023)

Reply: Thanks for your suggestion. We add the discussion about the rib-like pattern of basal friction in Section 5 of the revision: The rib-like pattern may also arise as compensating errors to other missing processes or uncertainties in datasets (Berends et al., 2023).

Ln 407: 'might be because'

Reply: Done.

**Ln 495: Did you also run with the same 1-year relaxation period?**

Reply: The 1-year relaxation was done before the inversion. We ran the Blatter-Pattyn model, restarting from the steady state result (i.e., inversion result) of the reference run using the full-Stokes model. We did not need do relaxation again using the Blatter-Pattyn model.

Ln 510: Cluster specific details are not necessary here in my opinion. Stating that Blatter Pattyn is 15 times faster would be sufficient.

Reply: Okay. We got your point. But it is 5 times faster, not 15 times faster. We change it to: Restarting from the steady state result using full-Stokes model in the reference run, the steady state simulation using Blatter-Pattyn model is 5 times faster than one timestep forward iteration using full-Stokes model with linear Weertman sliding parameterization.

521: Or, the structures arise as compensating errors to other missing processes or uncertainties in datasets, see my comment on Ln 393

Reply: We add the discussion about the rib-like pattern of basal friction in Section 5 of the revision: The rib-like pattern may arise as compensating errors to other missing processes or uncertainties in datasets (Berends et al., 2023).

Ln 571: This is not true, the rewriting was done first by (Brondex et al., 2017) Reply: We remove the words "give a novel way to".

**References**

Berends, C. J., Van De Wal, R. S., Van Den Akker, T., & Lipscomb, W. H. (2023). Compensating errors in inversions for subglacial bed roughness: same steady state, different dynamic response. *The Cryosphere*, *17*(4), 1585-1600. Brondex, J., Gagliardini, O., Gillet-Chaulet, F., & Durand, G. (2017). Sensitivity of grounding line dynamics to the choice of the friction law. *Journal of Glaciology*, *63*(241), 854-866.

Schoof, C. (2012). Marine ice sheet stability. Journal of Fluid Mechanics, 698, 62-72

**Referee 2**

I appreciate the additional work that the authors have carried out to address my review comments and to the further comments on those points from the editor.

I am satisfied that my concerns with the interpretation of the model results have been verified and then addressed in the updated manuscript. I think that the figures are improved from the first draft, and the points that needed additional discussion have been developed further.

When published, the differences in the modelled mass loss that the paper demonstrates due to changes in approximation scheme, sliding laws and melt rate forcings will be a valuable contribution to the ongoing discussions and developments in the field.

Reply: Thank you so much for your encouragement and helpful suggestions!

---

## Author Response (AR3)

Editor's comments are in blue, our reply in black, quotes in the revised manuscript in red.

Public justification (visible to the public if the article is accepted and published): Dear Ms. Yiliang Ma and Co-Authors,

thank you for submitting your revised manuscript. Please find below my remaining comments and a re-iteration of the comment of Reviewer 1 and myself on the necessity of a reference simulation to quantify model sensitivity. As soon as you have addressed the remaining comments and corrections we can proceed with the publication procedure.

With kind regards,

Johannes Sutter

L281-282: i think you are confusing Gt/yr and m/yr in Rignot et al 2013. Totten average melt rates in Rignot et al. 2013 are ~10 m/yr NOT 63.2 m/yr.

Reply: Yes. Sorry. We wrote the wrong unit. It is 63.2 Gt/yr, i.e., 10.5 m/yr. We corrected it.

L286-288: Same for Adusumilli et al., please be careful with the numbers you cite. The average basal melt rate for Adusumilli et al. 2020 is given as 11.5 m/a (see supplementary material table 1), thus very close to Rignot et al. and other publications. Please go through your manuscript carefully and double check any numbers you are citing from existing literature (e.g. line 327).

Reply: Yes, we agree. We double checked all the numbers we cite, and made sure the numbers are correct. We also find more papers and summarized earlier estimations based on satellite data, and updated it in the revision.

Flux gate calculations using satellite data have reported steady-state area-averaged basal melt rates of  $10.5 \pm 0.7$  m yr-1 (2003-2008; Rignot et al., 2013),  $9.89 \pm 1.92$  m yr-1 (2003-2009; Depoorter et al., 2013) and  $11.5\pm2.0$  m yr-1 (2010-2018; Adusumilli et al., 2020). Notably, Liu et al. (2015) used a similar method without assuming steady-state calving front, and derived a higher melt rate of  $17.9\pm1.2$  m yr-1 during the period 2003–2011. This elevated value aligns better with Rintoul et al.'s (2016) synoptic cavity water exchange estimates. In situ measurements by Vaňková et al. (2023) using autonomous phase-sensitive radar along grounding lines revealed large spatial melt variability, correlating with water column thickness gradients.

**References:**

Depoorter M. A., Bamber, J. L., Griggs, J. A., Lenaerts, J. T. M., Ligtenberg, S. R. M., van den Broeke, M. R., Moholdt, G., Calving fluxes and basal melt rates of Antarctic ice shelves. Nature, 502, 89-92, 2013.

Rintoul, S. R., Alessandro Silvano, Beatriz Pena-Molino, Esmee van Wijk, Mark Rosenberg, Jamin Stevens Greenbaum, Donald D. Blankenship, Ocean heat drives rapid basal melt of the Totten Ice Shelf. Sci. Adv. 2, e1601610, 2016.

Given the remarks above, I'm unsure whether you assumed too high melt rates in your melt parameterization in general. Thus, please quantify average/bulk melt rates in the sub-shelf melt rate fields shown in figure 3 so a comparison with the literature is made easier for the reader.

Reply: The area-averaged melt rates for our modelled sub-shelf melt rate with the maximum value of 20, 40 and 80 m yr-1 are 7.8, 14.0 and 26.6 m yr-1. Therefore, the area-averaged melt rates for modelled sub-shelf melt rate with the maximum value of 40 m yr-1 is closest to the satellite-based estimates in literature. We add in the revision Our model simulations for sub-shelf melt rate with the maximum value of 20, 40, 80 and 160 m yr-1 (Fig. 3) yield area-averaged melt rates of 7.8, 14.0, 26.6 and 51.6 m yr-1 respectively for the present day ice shelf geometry.

Since the modelled sub-shelf melt rate with the maximum value of 40 m yr-1 is closest to the satellite-based estimates (Section 3.3), the experiment  $C_{m_{40}}$  (which belongs to the Melt rate group) was selected as the control experiment to evaluate the model drift, and was done from 2015 to 2100.

L330 The sentence "and the sensitivity of Totten glacier dynamics to sub-shelf melt rate has been revealed in 35 years period." is unclear. What do you mean by "has been revealed in a 35 years period"?

Reply: The 35 years period is the simulation period 2015-2050 of Melt rate group experiments. We change the sentence to "and the sensitivity of Totten glacier dynamics to sub-shelf melt rate has been shown from 2015 to 2050"

Quantifying the drift of the reference-control run. You have dismissed the comments of the reviewer and my own comments regarding a quantification of the drift/trend in your reference simulation/control simulation. I agree with Reviewer 1 that such an assessment is still missing in your study, and that the simulation would be relatively easy to run in a reasonable amount of time. The comment on the trend of Reviewer 1 might be motivated by your Figure 9, in which even for C\_m0 the model responds with a linear and steady ice loss right from the beginning of the simulation implying model drift. This is fine but needs to be quantified and put into perspective with current trends in ice thickness changes in the region. If your thinning rates are higher(lower) than observations (e.g. Smith et al., 2020 or Nilsson et al., 2022 ESSD) suggest you can infer higher(lower) model sensitivity and quantify the trend.

Reply: We agree with the editor and the referee. After quantifying the area-averaged melt rates of our modelled sub-shelf melt rate with the maximum value of 20, 40, 80 and 160 m yr-1, we found the modelled sub-shelf melt rate with the maximum value of 40 m yr-1 is closest to the satellite-based estimates. Therefore, we took the simulation

C\_m40 using the modelled sub-shelf melt rate with the maximum value of 40 m yr-1 as the drift control experiment and extended it from 2050 to 2100.

We compared the modelled thinning rates from 2015 to 2020 with the observed (Nilsson et al., 2023). We found the observed and modelled annual mean thinning rates from 2015 to 2020 are -0.02 m/yr and -0.03 m/yr. We add in the revision that "In the drift control experiment C\_m40, the modelled annual mean thinning rate from 2015 to 2020 is -0.03 m yr $^{-1}$ , larger than the observed thinning rate of -0.02 m yr $^{-1}$  (Nilsson et al., 2023) , indicating higher model sensitivity to the present day climate forcing."

We quantify the model drift in the Results, section 4.2 of the revision as below. We add the  $C_{m40}$  result in Table 4. We also updated Figures 6, 7, 8 and 9 by adding the curve of  $C_{m40}$  result. Additionally, we polished the language in section 4.2 and adjusted some paragraphs according to the order in which the figures appeared.

Along the main trunk FL1 both the regularized Coulomb ( $C_m_{80}$ ) and linear Weertman ( $LW_m_{80}$ ) sliding parameterizations produce comparable grounding line retreat rates of 0.14 km yr-1 over 2015-2100 (Fig. 7). By 2050, this results in 8.08 km of retreat, increasing to 12.20 km by 2100. In contrast, with nonlinear Weertman sliding, retreats are 5.94 km by 2050 along FL1 followed by then stabilization (Fig. 6). Meanwhile, the drift control experiment ( $C_m_{40}$ ) shows a grounding line retreat of 8.08 km along FL1 by 2100.

Along FL2, the retreat rates differ significantly across parameterizations (Fig. 8): approximately 0.42 km yr-1 using linear Weertman (LW\_m80), 0.27 km yr-1 using regularised Coulomb (C\_m80) from 2015 to 2100, (Fig. 8), 0.04 km yr-1 from 2015 to 2050 followed by near-stabilization using non-linear Weertman sliding parameterization (NW\_m80). Along FL2, the grounding line retreats 12.09 km by 2050 and 35.85 km by 2100 using linear Weertman; 10.83 km by 2050 and 23.29 km by 2100 using regularised Coulomb; and only 1.32 km by 2050 followed by near-stabilization using nonlinear Weertman sliding parameterizations (Fig. 6). Meanwhile, the drift control experiment (C\_m40) shows a grounding line retreat of 12.33 km along FL2 by 2100.

The modelled grounded ice thinning rate in the reference run ( $C_{m80}$ ) near the projected 2100 grounding line is double that of the drift control run ( $C_{m40}$ ). Furthermore, surface ice velocity acceleration in  $C_{m80}$  at the projected 2100 grounding line position along the main trunk FL1 exceeds that in the drift control run by a factor of three over the 2015-2100 period (Table 4).

...The grounded area loss in LW $_{m80}$  and C $_{m80}$  experiments by 2100 significantly exceed the 3 413 km $^2$  from the drift control experiment....

The modelled VAF loss of the TG sub-basin from the Sliding parameterization group experiments is equivalent to global sea level rise of 5.67 mm, 5.48 mm and 3.29 mm over the period 2015-2050 (Fig. 9d) and 16.35 mm, 15.97 mm and 7.34 mm over the period 2015-2100 using linear Weertman, regularised Coulomb and non-linear Weertman sliding parameterizations, respectively, compared with 11.29 mm in the drift control experiment (Fig. 9c).

**Table 4.** Surface ice speed change,  $\Delta v$ , and ice thickness change rate,  $\Delta h$ , from the year 2015 to 2100 (2100 minus 2015) at the projected 2100 grounding line position and 15 km further upstream, along FL1 and FL2 from the drift control experiment and experiments in the sliding parameterization group.

|                       |                       | Along FL1 |                | Along FL2 |          |
|-----------------------|-----------------------|-----------|----------------|-----------|----------|
|                       | Experiment            | Grounding | 15 km Upstream | Grounding | 15 km    |
|                       |                       | Line 2100 |                | Line 2100 | Upstream |
| $\Delta v$            | C_m40 (drift control) | 87        | 21             | 210       | 145      |
| (m yr -1 ) | C_m80                 | 262       | 195            | 265       | 272      |
|                       | LW_m80                | 240       | 165            | 537       | 200      |
|                       | NW_m80                | 18        | -32            | -34       | -39      |
|                       | C_m40 (drift control) | -1.19     | -1.10          | -3.36     | -2.79    |
| $\Delta h$            | C_m80                 | -2.35     | -2.14          | -5.06     | -2.88    |
| (m yr -1 ) | LW_m80                | -2.65     | -2.29          | -6.42     | -2.83    |
|                       | NW_m80                | -0.91     | -0.59          | -0.79     | -0.70    |

**Reference:**

Nilsson, J., Gardner, A. S. & Paolo, F.: MEaSURES ITS\_LIVE Antarctic Grounded Ice Sheet Elevation Change. (NSIDC-0782, Version 1). [Data Set]. Boulder, Colorado USA. NASA National Snow and Ice Data Center Distributed Active Archive Center. https://doi.org/10.5067/L3LSVDZS15ZV, 2023.

**Figure 6.** Grounding line positions (coloured curves) at the year 2050 from experiments in the sliding parameterization group (a), at the year 2100 from experiments in the sliding parameterization group and drift control experiment (b), at the year 2050 from experiments in the melt rate group (c). The solid black line represents the initial grounding line position in the year 2015. The pink and purple solid lines are FL1 and FL2, respectively. The grey background shows

**the bed elevation.**

Figure 7. Surface ice speed (a, b), ice speed difference (c, d; surface minus basal), basal drag (e, f) ice thickness change (g, h), and ice-sheet profiles (i, j) along FL1 in the initial year 2015 (black solid line) and the year 2100 (coloured solid line) from experiments (Table 3) in the sliding parameterization group and drift control experiment (left column) and the year 2050 (coloured solid line) in the melt rate group (right column). Grounding line positions are marked with black vertical dashed line for the year 2015 and coloured vertical dashed line for the year 2100 in the Sliding parameterization group (left column) and the year 2050 (coloured solid line) in the melt rate group (right column). The figure is shaded to show the geometry of the TG along the flow line in 2015, where dark grey is the bedrock, light blue the ice shelf and dark blue seawater. The elevations are exaggerated by a factor of 25.

**Figure 8.** The same as Figure 7 but for FL2.

Figure 9. Grounded area change (a, b) and VAF change (c, d) of sliding parameterization group experiments and drift control experiment (a, c) over the period 2015-2100 and melt rate group experiments (b, d) over the period 2015-2050.

I want to point out here, that it is a usual and useful procedure in both model projection studies as well as sensitivity studies (such as yours) to define a baseline/reference/control simulation in which you simulate the evolution of the glacier/ice-sheet assuming no change in forcing (e.g. fixed CESM2 + present day ocean forcing e.g. similar to the ISMIP6 ocean reference field.). The response of your model against such a control scenario then provides an idea of the sensitivity of the model setup against which you can compare the perturbations in melt rate and SMB. This important aspect is still missing in your study. So far it is not possible to differentiate between already ongoing non-linearities in the ice sheet response vs. differences in the response triggered by different assumptions of basal sliding etc. Your study would be much more impactful if such an assessment where possible. As you employ an inversion in your model initialization, one useful simulation to quantify model behavior and sensitivity is to simulate (reg. Coulomb) the response of the catchment to present day constant forcing (with a bulk basal melt rate field optimally fitted to present day bulk melt rates, i.e. ~10 m/yr from Rignot et al., 2013 or Adusumilli et al., 2020) and compare the ice sheet thinning rates with satellite observations

So to re-iterate: please provide an additional reference run (reg. Coulomb) in which you prescribe a sub-shelf basal melt rate with an average rate similar to what the literature suggests (seems to be in the ballpark of 10 m/a) and present-day reference

SMB and surface air temperatures (e.g. in your case a historical or present day mean of CESM2). You can include this run in figure 9 as e.g. "present day reference simulation".

Reply: Agreed. So, we took the simulation  $C_{m_{40}}$  having the modelled sub-shelf melt rate with the maximum value of 40 m yr-1 which best matches the area-averaged observed rates, as the present-day reference experiment, and extended it from 2050 to 2100. We also compared the modelled and observed thinning rates from 2015 to 2020. We found the observed and modelled annual mean thinning rates from 2015 to 2020 are -0.02 m/yr and -0.03 m/yr, respectively. We also include the  $C_{m_{40}}$  result in Figure 9.

It would be also interesting to see how different the CESM2 SMB field looks like for present day compared to tailored regional climate models such as RACMO as often in ice sheet modelling studies the climate forcing is constructed with climate anomalies from a GCM added to a regional climate model to alleviate any regional model biases in globally tuned circulation models.

Additionally, please provide a time series (supplementary material) of CESM2 SSP5-85 SMB over the Totten catchment from 2015–2100, otherwise it is not possible to assess the influence of changing accumulation in the region. While you mention that CESM2 SMB does not vary much over the period from 2015–2100 it would be better to quantify this via the time series plot over both floating and grounded ice.

Reply: We double checked the SMB data we used. For the period 2015-2100, the applied SMB consists of repeated 1995-2030 CESM2 output under SSP585 scenario, which represents the present day atmospheric forcing conditions. We clarify this in the manuscript as

The surface mass balance (SMB) data applied for the period 2015-2100 consists of repeated 1995-2030 time-varying output of the Community Earth System Model Version 2 (CESM2) of the Coupled Model Intercomparison Project Phase 6 (CMIP6) under the SSP5-8.5 scenario at 32 km resolution (Nowicki et al., 2016; Fig. S1), which is close to the RACMO2.4p1 SMB data (Van Dalum et al., 2024).

We made a time series plot of SMB we used over the floating and grounded ice and compared it with RACMO2.4p1 as below. We add it in the supplementary material.

Figure S1. The SMB time series (blue curve) used for the period 2015-2100 in our experiments, consisting of repeated 1995-2030 CESM2 output, which is compared with the 1995-2023 SMB from RACMO2.4p1 data (orange dashed curve; Van Dalum et al., 2024).

**Reference:**

Van Dalum, C., Van de Berg, W. J., and Van den Broeke, M.: Monthly RACMO2.4p1 data for Antarctica (11 km) for SMB, SEB and near-surface variables (1979-2023), https://doi.org/10.5281/zenodo.14217232, Zenodo [data set], 2024.

**Please rephrase the second sentence in the abstract**

"It has the third highest annual ice discharge, 71.4±2.6 Gt yr-1, among East Antarctic outlet glaciers" as you cannot cite literature in the abstract and the number you give is probably from a specific study while there is a range of estimates. You could e.g. write something along the lines of

"It features very large discharge rates among the highest for East Antarctic outlet glaciers and has been losing mass over recent decades."

Reply: Thank you. We rephrase that sentence with the one you suggested.

---

## Author Response (AR4)

Editor's comments are in blue, our reply in black, quotes in the revised manuscript in red.

L394+: How come that both the C\_m80 and C\_m40 exhibit the exact same grl retreat until 2100? Maybe a mix up in the numbers? Please confirm whether these numbers are correct. -> "Along the main trunk FL1 both the regularized Coulomb (C\_m80) and linear Weertman (LW\_m80) sliding parameterizations produce comparable grounding line retreat rates of 0.14 km yr-1 over 2015-2100 (Fig. 7). By 2050, this results in 8.08 km of retreat, increasing to 12.20 km by 2100. In contrast, with nonlinear Weertman sliding, retreats are 5.94 km by 2050 along FL1 followed by then stabilization (Fig. 6). Meanwhile, the drift control experiment (C\_m40) shows a grounding line retreat of 8.08 km along FL1 by 2100."

Reply: Sorry, maybe the sentence is misleading. What we mean is that C\_m80 produces GL retreat of 8.08 km by 2050, and C\_m40 produce retreat of 8.08 km by 2100. To be clearer, we improved this sentence to be as below:

"Along the main trunk FL1 both the regularized Coulomb (C\_m80) and linear Weertman (LW\_m80) sliding parameterizations produce comparable grounding line retreat rates of 0.14 km yr-1 over 2015-2100 (Fig. 7). These rates corresponds to retreats of 8.08 km by 2050 and 12.20 km by 2100. In contrast, with nonlinear Weertman sliding, retreats are 5.94 km by 2050 along FL1 followed by then stabilization (Fig. 6). Meanwhile, the drift control experiment (C\_m40) shows a grounding line retreat of 8.08 km along FL1 by 2100."

Can you please structure the paragraph below (L400+) such that the annual retreat rates are followed by the integrated retreat until 2100 so the reader does not have to jump from one to the other through the parapgraph: -> approximately 0.42 km/yr (35.7 km until 2100) using linear Weertman (LW\_m80) etc.

I also was confused that you quantify 0.27km/yr or C\_m80 but 23.29 unti 2100. If I multiply 0.27 km/yr with 85 years I get 22.95km. The difference is small but should be zero if you compute the average retreat rate.

L400-406: "Along FL2, the retreat rates differ significantly across parameterizations (Fig. 8): approximately 0.42 km yr-1 using linear Weertman (LW\_m80), 0.27 km yr-1 using regularised Coulomb (C\_m80) from 2015 to 2100, 0.04 km yr-1 from 2015 to 2050 followed by near-stabilization using non-linear Weertman sliding parameterization (NW\_m80). Along FL2, the grounding line retreats 12.09 km by 2050 and 35.85 km by 2100 using linear Weertman; 10.83 km by 2050 and 23.29 km by 2100 using regularised Coulomb; and only 1.32 km by 2050 followed by near-stabilization using nonlinear Weertman sliding parameterizations (Fig. 6). Meanwhile, the drift control experiment (C\_m40) shows a grounding line retreat of 12.33 km along FL2 by 2100. "

Reply: Thanks for your suggestion. We change this paragraph to:

"Along FL2, the grounding line retreats 12.09 km by 2050 and 35.85 km by 2100 ( $\sim$ 0.42 km yr-1) using linear Weertman (LW\_m80); 10.83 km by 2050 and 23.29 km

by 2100 ( $\sim$ 0.27 km yr-1) using regularised Coulomb (C\_m80); and only 1.32 km by 2050 ( $\sim$ 0.04 km yr-1) followed by near-stabilization using nonlinear Weertman (NW\_m80) sliding parameterizations (Fig. 6)."

The accurate retreat rate for  $\sim$ 0.27 km/yr is 0.274 km/yr. So if we multiply 0.274 km/yr with 85 years, we get 23.29 km. But we reported the rates with two decimal places.

**L598 please correct the sentence.**

Reply: Corrected. We delete "as", and it becomes "Our simulations do not retreat past this slope before the year 2100."

Figure 5: I would suggest adding the grounding line position at the year 2100 of the drift\_ctrl experiment in figure 5b and 5c as well as in the transects (e.g. as a black or white line) so the relative positioning of the other runs with respect to the ctrl are accessible from the figure.

Reply: Okay, we improved Fig. 5 as the editor suggested.

**Figure 5.** Surface velocity at the beginning of the initial year 2015 (a) and the end of the year 2100 (b) in the reference run. The surface velocity difference (2100 minus 2015) is shown in (c). The grounding line positions in the year 2015, 2040, 2060, 2080 and 2100 in the reference run Cm80 are shown in (a-c). The grounding line position in the year 2100 in the drift control run Cm40 is shown in (b-g) with black lines. Pink and purple solid lines in (a) and (b) represent flowlines FL1 and FL2 as labelled. The solid color portions of the figures show the ice flow velocity (upper

colorbar) profiles along FL1 (**c**, **d**) and FL2 (**e**, **f**) in the reference run in the initial year 2015 (**d**, **f**) and the end year 2050 (**e**, **g**), with bedrock in dark grey and seawater in blue. The geometry change of TG is marked with colored solid lines for the years 2015, 2040, 2060, 2080 and 2100. The vertical elevations are exaggerated by a factor of 25.